# Evaluation of Development Potential of Ports in the Yangtze River Delta Using FAHP-Entropy Model

**Naixia Mou** [1,*] , **Chunying Wang** [1] , **Tengfei Yang** [2] and **Lingxian Zhang** [1]

1   College of Geomatics, Shandong University of Science and Technology, Qingdao 266590, China;
    wchunying@hotmail.com (C.W.); zhang_lingxian@163.com (L.Z.)
2   Institute of Remote Sensing and Digital Earth, Chinese Academy of Sciences, Beijing 100094, China;
    yangtf@radi.ac.cn
*   Correspondence: mounaixia@sdust.edu.cn

**Abstract:** Port development potential refers to the potential but unrealized status and capacity of ports, which can become a reality when external conditions permit. A correct analysis of port development potential helps to better formulate investment response plans and national development strategies, and finally achieve the sustainable development of the ports. Based on the Automatic Identification System (AIS) data, basic port data, hinterland city data, traffic network data, and relevant economic and policy data, we constructed an evaluation index system of port development potential, and evaluated the development potential of eight representative ports in the Yangtze River Delta port group of China with the methods of FAHP-entropy (FAHP—Fuzzy Analytical Hierarchy Process). The results show that: (1) The development potential of the port group in the Yangtze River Delta is positioned in the upper middle level; its development prospects are considerable, and other countries or ports could give priority of cooperation with it to maximize its benefits. (2) Port economy and policy are the primary core indicators affecting the development potential of ports, while per capita GDP (gross domestic product), number of berths, and port network status are the secondary core indicators affecting the development potential of ports. (3) Ports with larger development potential usually have one or more outstanding indicators, while the potential of ports with balanced development among all indicators is relatively weak.

**Keywords:** port development potential; FAHP-entropy; evaluation index system; location advantage; Yangtze River Delta; environmental initiatives

## 1. Introduction

More than 90% of the volume of trade in the world is completed by sea transportation [1]. Therefore, as a transportation node connecting land and sea, ports play an important role in the international trade [2]. It can be seen from the global urban development process that compared with inland cities, port cities and their surrounding areas are always the first to sprout, develop, and prosper, such as Singapore, New York, Hong Kong, and Shanghai. Shanghai Port is located in the Yangtze River Delta of China, and is the most important port along the 21st Century Maritime Silk Road. The city to which it belongs, Shanghai, is the financial center, shipping center, and one of the first batch of coastal open cities in China. Therefore, the development potential of Shanghai Port cannot be underestimated [3,4]. It should be noted that port development potential is different from port competitiveness. The former refers to the capability that may be realized in the future and then transformed into competitiveness, while the latter refers to the current static appearance and capability. Therefore, a quantitative analysis of port development potential is helpful not only to predict the port competitiveness in the future, but also to formulate corresponding development strategies according

to different development potentials of ports, so as to better drive the economic development of the surrounding areas.

Exploring and mastering the port development potential plays a key role in the sustainable development, transformation, and upgrading of ports. At present, the research methods of port development potential are mainly quantitative analysis and summary, and the research components include port operation efficiency [5,6], logistics development path [7], hinterland competitiveness [8], investment attraction [9], government policy support [10], and maritime competitive advantage [11,12]. Some of the research discusses the evaluation index system of port development potential. For example, Gao et al. [13] evaluated the development potential of Quanzhou Port from three aspects: government policy, investment trend, and investment risk management by fuzzy-AHP and ELECTRE III methods; Wan et al. [14] established the evaluation model of quantitative measurement of the development potential of inland ports in terms of the growth rate of port throughput, the Gross Domestic Product (GDP) growth rate of port cities, government support, and inland waterway classification, and used the Analytical Hierarchy Process (AHP) and D-S Reasoning method to dynamically evaluate the development potential of Wuhan Port in different years. The ranking of the highest potential container ports in 2017 was calculated to be Shanghai International Shipping Center of China [15], based on the six indicators; i.e., the growth rate of container throughput, the growth rate of GDP in the region where the port is located, the investment amount attracted by the port, the number of new routes, the natural conditions of the port (water depth, location), and the impact of government actions. To sum up, first of all, the current research on port development potential is mainly based on the port's own conditions, operation management, and political environment. However, each port is not isolated from the others in the complex maritime network. They interact and restrict each other through trade relations. Therefore, it is necessary to further consider such factors in the positioning of ports in the maritime network. However, at present, there is little research on the development potential based on the complexity of the maritime network, where the port is located, and the accessibility of the port hinterland transportation network. Therefore, this study introduced ships' Automatic Identification System (AIS) data, taking into account the trade relations between a port and other ports and the convenience of the port's external connection, and analyzed the data from a new perspective, contributing to a new and more detailed understanding of the port development potential. Besides, the frequently used methods of questionnaire and Delphi may have a certain subjectivity and one-sidedness, leading to the deviation of the prediction results. The combination of the methods FAHP and entropy was used to optimize and improve the qualitative analysis and quantitative analysis, and the integration of subjective evaluation and objective evaluation was to evaluate the port development potential more accurately.

On the basis of the existing research, by combining the complex network attributes of the maritime network and the big data of many sources, this paper constructs a new evaluation index system of port development potential. Case analyses of Shanghai Port, Ningbo-Zhoushan Port, Nanjing Port, Zhenjiang Port, Taicang Port, Nantong Port, Suzhou Port, and Jiangyin Port in China are displayed and used to explore the nature and law of port development potential. The combination of the research results and the current situation of port development can provide guidance for the sustainable development of ports.

## 2. Literature Review

Table 1 is the main literature review of this study, focusing on the evaluation indicators selected, the evaluation methods used, and the scope of the port involved in previous studies, which lead to the discussion and comparison of evaluation indicators, evaluation data, and evaluation methods in this paper.

**Table 1.** Summary of the main research on port development potential.

| References | Indicators | Methods | Area | Date |
|---|---|---|---|---|
| Gao et al. [13] | Government policy, investment trend, investment risk management | Fuzzy-AHP, ELECTRE III | Quanzhou Port | 2018 |
| Wan et al. [14] | Port throughput/GDP growth rate, government support, classification of inland waterways | AHP, D-S Reasoning | Wuhan Port | 2018 |
| Shanghai International Shipping Institute [15] | The growth rate of container throughput/GDP, the investment amount attracted by the port, the number of new routes, the natural conditions of the port, the impact of government actions | | Global container ports | 2017 |
| Feng et al. [16] | The growth rate of GDP/container throughput/cargo throughput/foreign trade import and export volume, port location advantage, port ecological environment | TOPSIS-entropy | Major container ports in China | 2017 |
| Peng et al. [17] | Density, trunk road, city scale, strategic channel, hub port, weighted degree centrality, weighted closeness centrality, weighted betweenness centrality | AHP-entropy | 99 ports along the Maritime Silk Road | 2018 |
| Pahl et al. [18] | Logistics demand and infrastructure | | Arctic ports | 2018 |
| Burling et al. [19] | The import and export goods | Potential growth model | Hedland Port | 2003 |

Note: TOPSIS—The Technique for Order of Preference by Similarity to Ideal Solution.

## 2.1. Relevant Studies on Evaluation Indicators

In order to determine port development potential, relevant studies have analyzed the evaluation indicators, including each port's infrastructure level, hinterland economic conditions, government support policies, and sea land competitive advantage from different perspectives. These studies are divided into four categories:

(1) Infrastructure operation. Pahl et al. [18] evaluated the development potential of Arctic ports from the perspective of logistics demand and infrastructure in combination with Arctic maritime economic activities; Burling et al. [19] established a potential growth model to study the development potential of Hedland Port based on the import and export goods of Hedland Port; Wiegmans et al. [20] tried to explain the relationship between the throughput of inland ports and the port development based on large-scale statistical data, and found that the development of ports mainly depended on the sufficient wharf size, diversified goods, and convenient transportation.

(2) Development of the port hinterlands. Min et al. [21] pointed out that the supply of port services exceeding the demand would lead to intensified port competition, and that accelerating the development of port hinterlands was a strong guarantee for government agencies and port enterprises to achieve a win-win situation. Notteboom et al. [22] found that port development had entered a new stage; i.e., ports had become the impact hub of their hinterlands. Ducruet et al. [23] analyzed the correlation between port hierarchies and city hierarchies from the perspective of relationships and networks, stating that although the correlation was weakening, ports with powerful cities still played an important role in the global maritime network. Van der Horst et al. [24] believed that the coordination of the hinterland transport chain was the main challenge for the future development of ports, and all departments should take corresponding incentive measures to improve the diversity of hinterland transport so as to improve the development potential of ports.

(3) Port policy and governance model. With the continuous development of the maritime market, the pressure on port capacity is increasing. Port efficiency and logistics management measures have become the key issues for market managers to consider [25]. With the refined logistics and outstanding policy advantages, free trade ports are major measures to open trade and promote development [26]. China officially opened the Shanghai Free Trade Zone (FTZ) in 2013,

which aims to reduce administrative intervention, realize tax preference, and relax investment restrictions [27]. In addition, the sustainable development policy is one of the main governance modes of the port. Schipper et al. [28] evaluated and guided the sustainable development of ports from three aspects: society, environment, and economy. Karimpour et al. [29] explored how ship waste discharge could increase its added value based on the closed-loop nature of circular economy, so as to promote the sustainable development of the Copenhagen-Malmö Port; Feng et al. [16], considering the port ecological environment index when assessing port development potential, pointed out that the core idea of the green port was not only to develop a port's economy, but also to take environmental and ecological protectionism into account, and to find a path of sustainable development.

(4) Sea and land competitive advantages of ports. Generally, scholars would use geographical location, hinterland economy, and port accessibility to reflect the location advantages of ports [30–32], thusly assessing the development potential of ports; however, Peng et al. [17] improved the assessment of port development potential through the network status from the perspectives of port location advantage and the global maritime transport network, and further studied the correlation between the development potential of a port and its position in the global maritime network.

## 2.2. Relevant Research on Evaluation Data

The maritime network has not only complex network characteristics, but complex geospatial characteristics as well. According to the specific geographical location of the port, the cargo transportation of the port and its position in the maritime network can be analyzed on micro and macro levels. Marine cargo transportation has diversity. According to the different modes of cargo transportation, the maritime network can be divided into container, bulk carrier, and tanker networks. The existing studies mainly focus on the spatial characteristics and evolution of the maritime container network, which mostly uses the shipping schedules provided by International, Barry Rogliano Salle (Alphaliner database), company's website, periodicals, and magazines [33,34]. The real-time data are difficult to obtain, so few people use the real-time positioning information of ships [35]—Lloyd's list, Lloyd's Register of Shipping, AIS data, etc. [36]. This paper uses AIS data from Lloyd's Register of Shipping, which contains three kinds of cargo transportation information: container, bulk carrier, and tanker, representing all real ships. In addition, Ducruet et al. [37] conducted some tests on the accuracy of Lloyd data, and found that it collected most of the marine transportation information and that the extracted data were enough to represent the global marine network flow. Lee et al. [38] indicated that the use of AIS data was beneficial to the public, enterprises, institutions, etc., and that the accessibility and availability of AIS data should be greatly enhanced. Therefore, the AIS data used in this paper are conducive to a more comprehensive discussion of the maritime network, and thus can provide more accurate optimization suggestions for the sustainable development of the port.

## 2.3. Relevant Studies on Evaluation Methods

Relevant studies use different methods to explore the port development potential, such as comparative analysis, cluster analysis, and combinations of various methods [39]. Port development potential itself is a fuzzy concept of multiple levels, multiple factors, and multiple variables, and when the index value cannot be clearly indicated, fuzzy evaluation is considered to be the most appropriate method [40]. In fact, fuzzy evaluation is usually applied jointly with other evaluation methods [41,42]. The representative joint method is the AHP method, which can not only sort out the indexes hierarchically, but make a scientific and reasonable quantitative evaluation of the information contained in the index as well; but the construction of its judgment matrix must still be interfered by subjective factors [43]. The entropy method can help to optimize the construction of the judgment matrix and weaken the influence of subjective factors. Therefore, the combination of fuzzy, AHP, and entropy methods can not only reduce the subjective impact, but also consider the characteristics of all indicators,

making the evaluation results more accurate and in line with reality, so an FAHP-entropy model was used in this study. The advantages and disadvantages of entropy, AHP, and fuzzy are shown in Table 2.

**Table 2.** The advantages and disadvantages of entropy, AHP and fuzzy methods.

| Method | Advantage | Disadvantage |
|---|---|---|
| Entropy | Objective, used in any process of weight determination | Only used in the process of weight determination, limited problem solving |
| AHP | Systematic, concise and practical | Many qualitative components are not convincing, difficult to determine the weight when there are too many indicators |
| Fuzzy | Quantifiable evaluation of fuzzy concepts, making qualitative problems quantifiable | Complex calculation, subjective determination of index membership function |

Entropy, AHP, fuzzy, and their combinations are often used in the research of development potential. The same sample data were used to test the evaluation results of the FAHP-entropy model and other three models, as is shown in Appendix A (Table A1). It can be seen that the compatibility of the FAHP-entropy model is the highest among the four models, and the representativeness of the computing results is better than what resulted from the other three models. In fact, compared with AHP-entropy, FAHP-entropy has the advantages of being able to blur the indicators that cannot be quantified; compared with FAHP, it has the advantages of being able to weaken the influence of subjective factors and offer objective weight; compared with entropy, it has the advantage of being able to systematically and hierarchically analyze the impact degree of each indicator on the results.

In view of all that, this study simulated the global maritime network based on AIS data, analyzed the position and importance of each port in the maritime network by using the complex network method, and combined the qualitative and quantitative indicators, considering factors such as port infrastructure, hinterland economy, hinterland traffic, policy support, and the environmental protection concept. Moreover, four primary indicators—the port shipping level, port economy and policy, port seaward location advantage, and port landward location advantage, and 14 secondary indicators, were selected to constitute the evaluation index system. The FAHP-entropy method was used to analyze eight representative ports in the Yangtze River Delta port group of China. The main contributions of this study are as follows: firstly, the degree of influence of each factor on the development potential of the port was determined, and a scientific evaluation index system is proposed herein, which was combined with the concept of green ports to provide suggestions for the sustainable development of ports; secondly, from the perspective of a complex network, the maritime network of containers, bulk carriers, and tankers was comprehensively considered, and the position of each port in the global maritime network was determined; thirdly, the subjective issue of constructing a judgment matrix in AHP method and membership function in the fuzzy method were improved with the objective basis provided by entropy weight.

## 3. Data

With Shanghai as the core, the Yangtze River Delta port group is composed of 24 ports in Jiangsu Province, Zhejiang Province, and part of Anhui Province. It is a region with one the most dynamic economies, the highest degree of openness, the strongest capacity for innovation, and the largest foreign population in China—and even in the world. It is an important intersection of the 21st century Maritime Silk Road and China's Yangtze River Economic Belt with frequent domestic and foreign trade, and a world-class port group radiating the Asian-Pacific region and leading the world [44]. According to the port throughput and hinterland development, we selected eight representative ports in the Yangtze River Delta as the research object; namely, Shanghai Port, Ningbo-Zhoushan Port, Nanjing Port, Zhenjiang Port, Taicang Port, Nantong Port, Suzhou Port, and Jiangyin Port. The distribution of research ports is shown in Figure 1, and the research data source is shown in Table 3. As a new navigation instrument, AIS aims to avoid ship collision and ensure maritime safety [45]. Through the

data collection of AIS, the monitoring of global ships can be realized, including the dynamic and static information, such as ship type, ship position, ship direction, ship speed, and load. Besides, the use of AIS data is no longer limited to maritime applications because the AIS contains a lot of valuable information for the port and shipping industry, and the examination of AIS data can provide a scientific basis for the business decision-making of the port and shipping industry [46]. In addition, compared with traditional data, AIS data can better reflect the real situation of maritime transportation, because it contains the bulk carrier and tanker data that cannot be covered by sailing schedules [47]. Therefore, the AIS data in this study provided all cargo ship information, which can more accurately reflect the position and importance of each port in the maritime network.

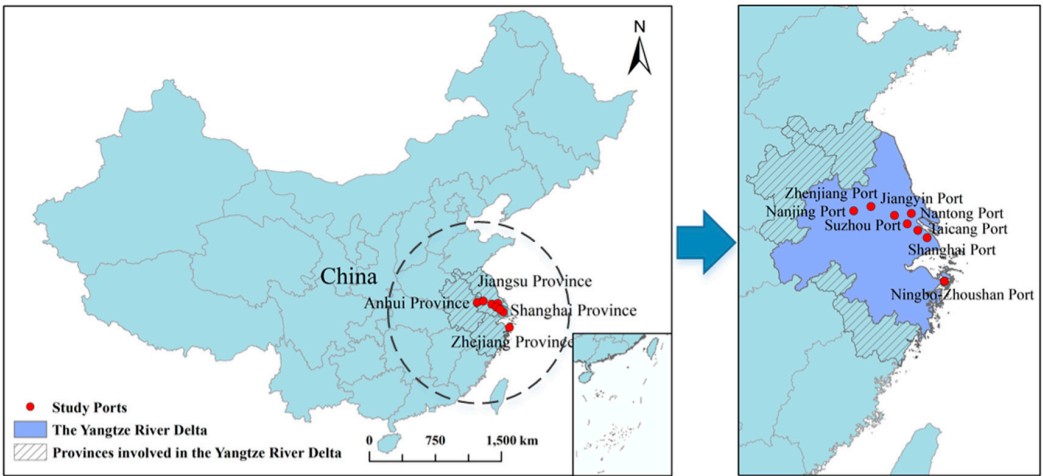

**Figure 1.** The ports in question.

**Table 3.** The data sources.

| Data | Data Sources |
|---|---|
| Hinterland GDP data, hinterland area, investment amount | National Data (http://data.stats.gov.cn/) [48], port annual report [49] |
| Number of routes, AIS data | HiFleet Ltd. (http://www.hifleet.com) [50] |
| City information, road network information | OpenStreetMap (http://www.openstreetmap.org/) [51] |
| Port throughput, terminal berths, related policies | China Port (http://www.port.org.cn) [52], provincial statistical bulletin in 2014, 2015 [53] |
| The distribution of strategic hubs and strategic channels | World Port Index (http://www.nga.mil/) [54], and reference [55] |

All data are from January 1, 2014 to March 31, 2015.

## 4. Methodology

In this paper, a FAHP-entropy model is proposed by integrating the methods of entropy, AHP, and Fuzzy, which can to some extent avoid the fluctuation of evaluation accuracy caused by the interference of subjective factors.

Figure 2 presents the methodological framework of port development potential evaluation, which consists of eight steps:

Step 1. Determining the evaluation methods.

Step 2. Determining the evaluation index system.

Step 3. Determining the score of each port on each index and calculating the weight of each index by the entropy method.

Step 4. Using entropy weight to help construct the judgment matrix of AHP.

Step 5. Using a judgment matrix to obtain the results of hierarchical single sorting of each index and each port.

Step 6. Using the entropy weight again, and the results of hierarchical single sorting, to establish the membership function of each port in the hierarchical fuzzy subset.

Step 7. Using a weighted average M (⊕) fuzzy composition operator, membership function, and entropy weight to obtain the port development potential.

Step 8. Testing the compatibility degree of each weighting method.

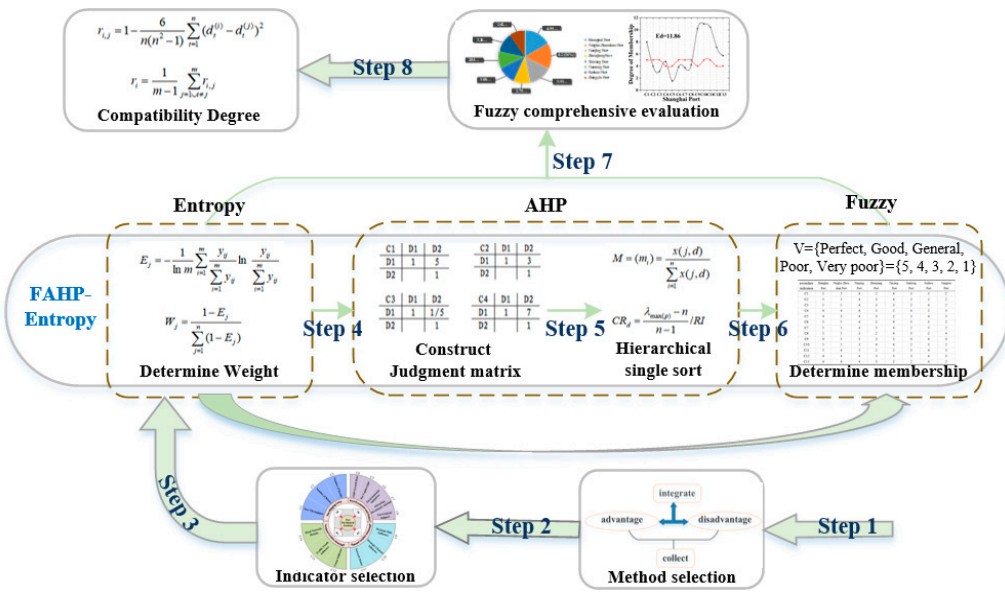

**Figure 2.** The methodological framework of port development potential evaluation.

*4.1. Construction of Evaluation Index System*

Referring to a large number of documents on port competitiveness evaluation and development potential evaluation, and considering the basic connotations of port development potential, a theoretical framework, and ports' positions in the maritime network, the port development potential evaluation index system was constructed, which includes four primary indicators—port shipping level, port economy and policy, port seaward location advantage, and port landward location advantage (B1–B4), and 14 secondary indicators (C1–C14), as is shown in Figure 3.

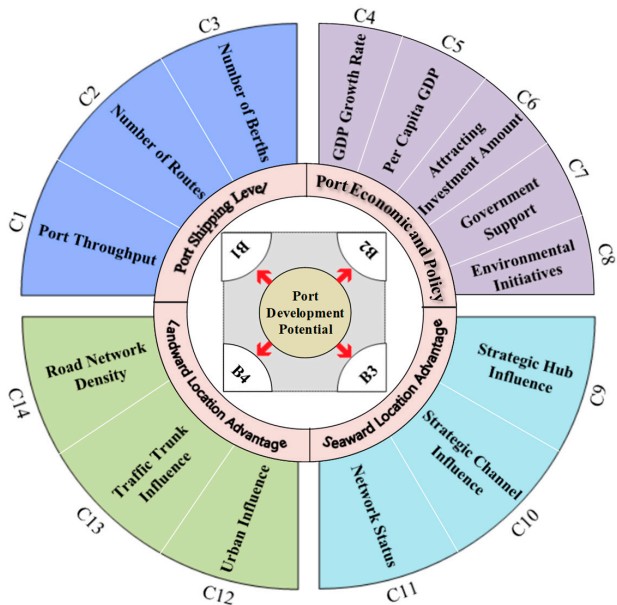

**Figure 3.** Port development potential evaluation index system.

(1) B1: Port shipping level refers to the level of production and operation reached by the port under its natural conditions.

C1: Port throughput (PT) refers to the quantity of goods that has been loaded and unloaded at the port over a period of time, reflecting the ability of the port to produce and operate activities.

C2: Number of routes (NR) refers to the existence of goods or passenger transportation between the port and other ports. The more the routes are, the more advantageous the port is.

C3: Number of berths (NB) refers to the maximum number of ships that can be docked at the same time. An insufficient number of berths causes the ships to queue, lowering the transportation efficiency, which is not conducive to port development.

(2) B2: Port economy and policy refer to the impact of the economic level of the hinterland of the port, and related policies implemented by the government impacting the development of the port.

C4: GDP growth rate (GDPGR) refers to the economic development trend of the hinterland of the port and the economic support ability for the future development of the port. The calculation model is:

$$C4 = \frac{G_1 - G_2}{G_2} \times 100\%, \tag{1}$$

where $G_1$ represents the total GDP of the current hinterland, and $G_2$ represents the total GDP of the past hinterland.

C5: Per capita GDP (PCGDP) refers to the embodiment of the overall living standard of the people in the hinterland of the port, calculated by dividing the total GDP by the population.

C6: The attracting investment amount (AIA) can provide sufficient financial support for port development, and the amount of investment attraction also reflects the development potential of the port.

C7: Government support (GS) mainly includes the economic support and policy inclinations provided by the government and the state; certain ports will give priority to development, and rapid development at that, thereby driving the development of port cities and their surrounding cities.

C8: Environmental initiatives (EI) mainly include the port environmental protection management policies. This index is closely related to the sustainable development of the port.

(3) B3: Port seaward location advantage refers to the current port development status formed by the port's competitive advantage based on its sea.

C9: Strategic hub influence (SHI) refers to the importance and strategic position of the port in the global shipping network. As shown in Appendix B (Figure A2), the ports with high influence are mainly located in strategic areas, such as Singapore Port. The hub port plays a key role in the connectivity of the global maritime network [56]. Through the identification model of global strategic shipping pivots, 44 strategic hubs, seven strategic channels, and three strategic sea areas were identified [55]. The calculation model is:

$$C9 = \sum_{j=1}^{n} \left( k d_{ij} + \omega_j \right), \tag{2}$$

where $C9$ represents the strategic hub influence of port $i$, $k$ represents the attenuation coefficient ($k < 0$), $d_{ij}$ represents the shortest distance between port $i$ and hub port $j$, and $\omega_j$ represents the weights of ports of different levels.

C10: Strategic channel influence (SCI) refers to the importance of a certain area for gathering global ship traffic, as shown in Appendix B (Figure A3), and the score is based on distance.

C11: Network status (NS) refers to the structural characteristics of the maritime network. Based on AIS data, this paper explores the influence and control of each port on the global maritime network pattern according to degree index, node number, and node tightness in the complex network [57]. The calculation model is:

$$C_d(i) = \sum_{i \neq j \in N} \theta_{ij} \tag{3}$$

$$C_b(i) = \sum_{j \neq m \in N} \frac{\sigma_{jm}(i)}{\sigma_{jm}} \tag{4}$$

$$C_c(i) = \sum_{i \neq j \in N} d_{ij} \tag{5}$$

$$C11 = C_d(i) \times \omega_1 + C_b(i) \times \omega_2 + C_c(i) \times \omega_3, \tag{6}$$

where $C11$ represents the network status of port $i$, and $C_d(i)$, $C_b(i)$, and $C_c(i)$ represent the degree index, node number, and node index of port $i$, respectively. $\omega_1$, $\omega_2$, and $\omega_3$ represent the weight of each indicator, respectively; $i$, $j$, and $m$ indicates different ports in the collection $N$; and $\theta_{ij}$ indicates the port connection. If connected, $\theta_{ij} = 1$; otherwise, $\theta_{ij} = 0$. $\sigma_{jm}$ indicates the number of the shortest paths between two ports; $\sigma_{jm}(i)$ indicates the number of ports $i$ passing through the shortest path between port $j$ and port $m$; and $d_{ij}$ indicates the shortest distance between the two ports.

(4) B4: Port landward location advantage refers to the current port development status formed by the port's competitive advantage based on its hinterland.

C12: Urban influence (UI) refers to the support capacity of the hinterland to the port, through the development level of the hinterland of the port and the distance between the port and the city. The calculation model is:

$$C12 = G_i \, exp\left(-U_{d_{ij}}\right), \tag{7}$$

where $C12$ represents the urban influence of port $i$, $G_i$ represents the hinterland GDP of port $i$, and $U_{d_{ij}}$ indicates the shortest distance between port $i$ and city $j$.

C13: Traffic trunk influence (TTI) refers to the supportability of the hinterland traffic trunk line in regard to the port through traffic accessibility. The calculation model is:

$$C13 = \sum_{j=1}^{n}\left(kd_{ij} + \omega_j\right), \tag{8}$$

where $C13$ represents the traffic trunk influence of port $i$, $k$ represents the attenuation coefficient ($k < 0$), and $d_{ij}$ represents the shortest distance between port $i$ and road $j$. The traffic facilities in this paper include railway stations, trunk roads, highway entrances, and airports. $\omega_j$ represents the weights of different facilities, as shown in Appendix B (Table A3).

C14: Road network density (RND) refers to the calculation of the comprehensive road network density of railways, trunk roads, and highways in the hinterland of the port. The calculation model is:

$$C14 = \sum_{j=1}^{n} \omega_j \frac{L_j}{A_i}, \tag{9}$$

where $C14$ represents the density of the hinterland road network of port $i$, $\omega_j$ represents the weight of each road, $L_j$ represents the total length of each road in the hinterland, and $A_i$ represents the area of the hinterland.

Among the above C1–C14 indicators, C7, C8, and C10 are qualitative indicators, while the others are quantitative indicators. The grades of C7 and C8 are described by "perfect," "good," "average," "poor," and "very poor" with corresponding scores of 5, 4, 3, 2, and 1. The grades of C10 are 1, [0.5, 1], [0, 0.5) respectively for "located in," "close to," and "far from " strategic channels, and each specific value is determined by the shortest distance from the strategic channel. The evaluation criteria for each level of qualitative indicators are listed based on the reference [58] and expert recommendations, as is shown in Table 4, and the quantitative indicators can be calculated by Table 3 and Equations (1)–(9). Appendix B (Table A2) further explains the data, data sources, and processing steps required for each indicator. Besides, it should be noticed that all qualitative indicators and quantitative indicators need to be normalized after obtaining scores.

**Table 4.** Standards of each grade for qualitative indicators.

| Qualitative Indicators | | C7 Government Support | C8 Environmental Initiatives |
|---|---|---|---|
| **Definition of each grade** | Perfect (5) | Sound port management regulations, and strong support from government on both financial and political aspects. | Government attaches great importance to environmental initiatives, promulgates a lot of port environmental protection management policies, and gives subsidies incentives. |
| | Good (4) | Unified regulation on port management, and support from the government on both financial and political aspects. | Government attaches great importance to environmental initiatives, promulgates some port environmental protection management policies. |
| | Average (3) | Unified regulation on port management, and financial support from the government. | Government attaches great importance to environmental initiatives, few policies promulgated, but specific actions taken. |
| | Poor (2) | Unified regulation on port management, or support from the government on both financial and political aspects. | Government pays attention to environmental initiatives and tries to promulgates some port environmental protection management policies. |
| | Very poor (1) | No official regulation on port management, no financial support from the government. | Government does not pay much attention to environmental protection, and has not promulgated any policies. |
| | | **C10 Strategic channel influence** | |
| | 1 | In the position of strategic channel | |
| | [0.5, 1) | Close to strategic channel | |
| | [0, 0.5) | Away from strategic channel | |

*4.2. Calculation of the Evaluation Index Weight*

The weight of the evaluation indicators is determined by entropy, and the specific processes are as follows:

(1) Data normalization. The evaluation indicators mainly include two types: benefit indicators (the bigger the data, the better) and cost indicators (the smaller the data, the better). Because different indicators have differences in dimension and magnitude, data need to be normalized for unified standard. The evaluation indicators in this paper are all benefit indicators. Therefore, for $m$ research areas, the normalization method of the initial matrix of $n$ evaluation indicators are as follows:

$$
\begin{aligned}
X &= \left(x_{ij}\right)_{m \times n} \quad i = 1,\ 2,\ \dots,\ m;\ j = 1,\ 2,\ \dots,\ n \\
Y &= \left(y_{ij}\right)_{m \times n} \quad i = 1,\ 2,\ \dots,\ m;\ j = 1,\ 2,\ \dots,\ n \\
y_{ij} &= \frac{max\left(x_j\right) - x_{ij}}{max\left(x_j\right) - min\left(x_j\right)},
\end{aligned}
\tag{10}
$$

where $X$ represents the initial data matrix and $Y$ represents the normalized benefit data matrix.

(2) Calculating the information entropy and weight of each indicator. Entropy, first introduced to the information theory by the mathematician C.E. Shannon, has been widely used in such fields as engineering technology and social science. According to the interpretation of information theory, information is a measure of the order of a system, while entropy is a measure of the disorder of a system. Therefore, the smaller the information entropy of an indicator, the more information it provides, the more important its status, and the higher the weight; vice versa. The specific empowerment process is as follows:

$$
p_{ij} = \frac{y_{ij}}{\sum_{i=1}^{m} y_{ij}}
\tag{11}
$$

$$
E_j = -\frac{1}{ln\ m} \sum_{i=1}^{m} p_{ij}\ ln\ p_{ij}
\tag{12}
$$

$$
W_j = \frac{1 - E_j}{\sum_{j=1}^{n}\left(1 - E_j\right)},
\tag{13}
$$

where $p_{ij}$ represents the proportion of the $i$-th port under the $j$-th indicator, $E_j$ represents the entropy value of the $j$-th indicator, and $W_j$ represents the entropy weight of the $j$-th indicator.

## 4.3. Calculation of Port Development Potential

(1) Construct a judgment matrix *A* that reflects the relationship between the two indicators. Among them $A = (a_{ij})_{n \times n}$, where *A* indicates the relative importance of all indicators of the hierarchy to a certain indicator of the previous level (relative importance is determined by the weight obtained by the above-mentioned entropy weight method). The element $a_{ij}$ in the matrix is given by Santy's 1–9 scale method, as is shown in Table 5.

**Table 5.** The scaling method.

| Scale Value | Relative Importance of the Two Indicators |
|:---:|:---:|
| 1 | Indicator *i* is as important as indicator *j* |
| 3 | Indicator *i* is slightly more important than indicator *j* |
| 5 | Indicator *i* is medium more important than indicator *j* |
| 7 | Indicator *i* is very more important than indicator *j* |
| 9 | Indicator *i* is absolutely more important than indicator *j* |
| 2, 4, 6, 8 | Intermediate value of the above importance |

(2) The results of hierarchical single sorting should then be used to determine the indicator's membership degree to the hierarchical fuzzy subsets. Hierarchical single sorting means to calculate the weight ranking of each port under a single indicator, and to judge the logical correctness of the ranking through the consistency test. Otherwise, there will be contradictions in the ranking, and the judgment matrix *A* needs to be re-adjusted. Divide each indicator into five levels at equal intervals, with level set V = {v1, v2, v3, v4, v5} = {perfect, good, average, poor, very poor} = {5, 4, 3, 2, 1}. A membership function was established based on the above weight values. Using a weighted average M (⊕) fuzzy composition operator, the fuzzy set of the upper-level index is obtained through the index weights of various levels and corresponding membership function, and the evaluation level of the final port development potential is obtained according to the principle of maximum membership degree.

$$M = (m_i) = \frac{x(j,d)}{\sum_{i=1}^{n} x(j,d)} \quad i, j = 1, 2, \ldots, n \tag{14}$$

$$CI = \frac{\mu_{max(p)} - n}{n - 1} \tag{15}$$

$$CR_d = \frac{CI}{RI} \tag{16}$$

$$A_i = R_i \times U_i, \tag{17}$$

where *M* represents the weight of each port under a single indicator, *x* represents the eigenvector matrix of the judgment matrix *A*, $\mu_{max(p)}$ represents the maximum eigenvalue of the judgment matrix *A*, and *d* represents the column of the largest eigenvalue. *CI* represents a single-order consistency indicators; $CR_d$ represents the single-sequence consistency ratio; *RI* represents the average random consistency index, obtained by querying the *RI* value reference table; and *n* is the number of pairs of comparison indicators. In general, when $CR_d < 0.1$, the hierarchical single sorting result is considered to have satisfactory consistency. $A_i$ indicates the degree of membership of each evaluation level corresponding to the higher level index, and $U_i$ indicates the weight of each index.

## 4.4. Compatibility Degree Test

In order to visually analyze the robustness difference between FAHP-entropy model (Sections 4.2 and 4.3) and other models in evaluating the development potential of the port, a compatibility test should be adopted. Model compatibility degree refers to arithmetic mean of the rank correlation coefficient between the two models, and the reliability and practical value of the model tested by

the Spearman correlation coefficient; i.e., a compatibility test [59]. The greater the compatibility is, the better the model behaves in the aspect of robustness. The calculation model is:

$$r_{i,j} = 1 - \frac{6}{n(n^2-1)} \sum_{t=1}^{n} \left( d_t^{(i)} - d_t^{(j)} \right)^2 \; i, j \; = \; 1, 2, \ldots, m \tag{18}$$

$$r_i = \frac{1}{m-1} \sum_{j=1 \cup i \neq j}^{m} r_{i,j} \; i, j \; = \; 1, 2, \ldots, m, \tag{19}$$

where $r_{i,j}$ denotes the Spearman rank correlation coefficient between the two models of $I$ and $j$; $n$ denotes the number of evaluation objects; m denotes the number of models; $d_t^{(i)}$ and $d_t^{(j)}$ denote the rankings of evaluation object $t$ in $i$ and $j$ models respectively; and $r_i$ indicates the compatibility of model $i$.

## 5. Results and Discussion

### 5.1. Results of Port Development Potential

Firstly, the scores of 14 indicators of eight ports were obtained through Table 3, Table 4, and Equations (1)–(9), and then the weight of each index was calculated according to Equations (10)–(13), as is shown in Table 6. From the results of index weight, it can be seen that the impact of each indicator on port development potential is: port economic and policy (0.425), port shipping level (0.246), port landward location advantage (0.196), and port seaward location advantage (0.133).

**Table 6.** The weights of evaluation indexes.

| Primary Indicators | Weights | Secondary Indicators | Weights | Comprehensive Weights |
|---|---|---|---|---|
| Port shipping level | 0.246 | Port throughput | 0.179 | 0.044 |
| | | Number of routes | 0.362 | 0.089 |
| | | Number of berths | 0.459 | 0.113 |
| Port economic and policy | 0.425 | GDP growth rate | 0.131 | 0.056 |
| | | Per capita GDP | 0.408 | 0.173 |
| | | Attracting investment amount | 0.161 | 0.068 |
| | | Governmental support | 0.167 | 0.071 |
| | | Environmental initiatives | 0.133 | 0.057 |
| Port seaward location advantage | 0.133 | Strategic hub influence | 0.186 | 0.025 |
| | | Strategic channel influence | 0.119 | 0.016 |
| | | Network status | 0.695 | 0.092 |
| Port landward location advantage | 0.196 | Urban influence | 0.233 | 0.046 |
| | | Traffic trunk influence | 0.342 | 0.067 |
| | | Road network density | 0.425 | 0.083 |

According to Equations (14)–(17), the weights of all the indexes determined by the entropy model were used to establish a judgment matrix reflecting the relationship between the two indicators. The final judgment matrix was determined by verifying the consistency of sorting logic, thusly obtaining the indicators and the levels of each port. Single sorting results as shown below (Table 7).

According to the entropy weights of Table 6 and the hierarchical single sorting results of Table 7, the membership function of each port in the hierarchical fuzzy subset was constructed, and the entropy weight was further combined with the entropy weight to obtain the evaluation level of the primary indicators and secondary indicators, and the development potential level of each port.

**Table 7.** Hierarchical single sorting results of port evaluations.

| C1–C14. | Shanghai Port | Ningbo–Zhoushan Port | Nanjing Port | Zhenjiang Port | Taicang Port | Nantong Port | Suzhou Port | Jiangyin Port |
|---|---|---|---|---|---|---|---|---|
| C1 PT | 0.284 | 0.357 | 0.118 | 0.027 | 0.120 | 0.032 | 0.034 | 0.028 |
| C2 NR | 0.363 | 0.313 | 0.102 | 0.042 | 0.065 | 0.043 | 0.040 | 0.032 |
| C3 NB | 0.390 | 0.308 | 0.072 | 0.046 | 0.045 | 0.035 | 0.064 | 0.040 |
| C4 GDPGR | 0.172 | 0.143 | 0.203 | 0.099 | 0.041 | 0.127 | 0.094 | 0.121 |
| C5 PCGDP | 0.170 | 0.140 | 0.161 | 0.092 | 0.022 | 0.088 | 0.162 | 0.165 |
| C6 AIA | 0.236 | 0.165 | 0.135 | 0.054 | 0.156 | 0.079 | 0.097 | 0.078 |
| C7 GS | 0.255 | 0.147 | 0.139 | 0.089 | 0.124 | 0.076 | 0.107 | 0.063 |
| C8 EI | 0.176 | 0.133 | 0.115 | 0.100 | 0.137 | 0.104 | 0.124 | 0.111 |
| C9 SHI | 0.310 | 0.148 | 0.094 | 0.068 | 0.117 | 0.087 | 0.072 | 0.104 |
| C10 SCI | 0.217 | 0.054 | 0.026 | 0.026 | 0.026 | 0.217 | 0.217 | 0.217 |
| C11 NS | 0.292 | 0.169 | 0.087 | 0.103 | 0.125 | 0.075 | 0.095 | 0.054 |
| C12 UI | 0.215 | 0.124 | 0.150 | 0.102 | 0.074 | 0.106 | 0.115 | 0.114 |
| C13 TTI | 0.208 | 0.121 | 0.197 | 0.083 | 0.080 | 0.108 | 0.127 | 0.076 |
| C14 RND | 0.238 | 0.146 | 0.154 | 0.112 | 0.097 | 0.081 | 0.132 | 0.040 |

Note: PT indicates port throughput; NR indicates number of routes; NB indicates number of berths; GDPGR indicates GDP growth rate; PCGDP indicates per capita GDP; AIA indicates the attracting investment amount; GS indicates government support; EI indicates environmental initiatives; SHI indicates strategic hub influence; SCI indicates strategic channel influence; NS indicates network status; UI indicates urban influence; TTI indicates traffic trunk influence; RND indicates road network density.

*5.2. Analysis of Evaluation Results*

The results of port development potential are shown in Table 8, and the ranking is in the order: Shanghai Port, Ningbo-Zhoushan Port, Nanjing Port, Suzhou Port, Taicang Port, Zhenjiang Port, Jiangyin Port, and Nantong Port. Figure 4 shows the rankings of the eight ports under the four primary indicators of port shipping level (B1), port economy and policy (B2), port seaward location advantage (B3), and port landward location advantage (B4). Combined with Table 8 and Figure 4, it can be seen that Shanghai Port and Ningbo-Zhoushan Port occupy the top two, mainly because they are superior to other ports in terms of shipping level and sea location advantage. In addition, the development potential of each port presents gradient distribution, and the restriction conditions of each port are different. Shanghai Port, Nanjing Port, Suzhou Port, and Nantong Port are mainly limited by the competition for the sea supply, which indicates that the positions of these ports in the global maritime network can still be improved, while other conditions of the ports are enough to further support the fierce competition at sea; Ningbo-Zhoushan Port and Zhenjiang Port are mainly limited by the economic level in the hinterland and the lack of relevant policies issued by the government, which do not play a good supporting role in the future development of the ports; Jiangyin Port, mainly due to the constraints of its own shipping level, should appropriately increase the number of berths, expand trade cooperation, and promote its own development; Taicang Port is mainly restricted by the traffic level in the hinterland, so it should continue to improve the inland transportation network, and provide efficient and all-round logistics services, such as loading and unloading, warehousing, processing, collection, and distribution, so as to expand its development potential.

**Table 8.** Overall evaluation results of port development potential.

| Port | Shanghai Port | Ningbo–Zhoushan Port | Nanjing Port | Zhenjiang Port | Taicang Port | Nantong Port | Suzhou Port | Jiangyin Port |
|---|---|---|---|---|---|---|---|---|
| Evaluation score | 4.55 | 4.25 | 3.91 | 2.75 | 3.00 | 2.84 | 3.28 | 2.62 |
| Evaluation level | Perfect | Perfect | Good | Average | Good | Average | Good | Average |

It was found that in the indicator of port economy and policy, the gap between the maximum value and the minimum value was the largest, and the data distribution was relatively scattered, which shows that the eight representative ports in the Yangtze River Delta port group have a large gap in hinterland economic level and government policy support; while in the indicator of port seaward location advantage, the gap between the maximum value and the minimum value was the smallest,

and the data distribution was relatively centralized, which indicates that the eight representative ports in the Yangtze River Delta port group have a small gap in their positions in the global shipping network, and the competition of maritime supply is less than the competition of land supply. According to the weighting results of primary indicators, the port economy and policy (0.425) is the key factor for determining the development potential of the port. Due to the large gap between the port economy and policy in the Yangtze River Delta, the gap between the development potential is also large. The development potential of Ningbo-Zhoushan Port is still far behind that of Shanghai Port. This is mainly due to the constraints of its economy and the lack of policies of the Ningbo-Zhoushan Port. If Ningbo-Zhoushan Port can improve the economic level of the hinterland and formulate favorable policies in a short period of time, the development potential of Ningbo-Zhoushan Port is likely to catch up with that of Shanghai Port, because Ningbo-Zhoushan Port is basically the same as Shanghai Port in terms of natural conditions and competition for land and sea resources. In the future, Ningbo-Zhoushan Port will become an important global strategic hub.

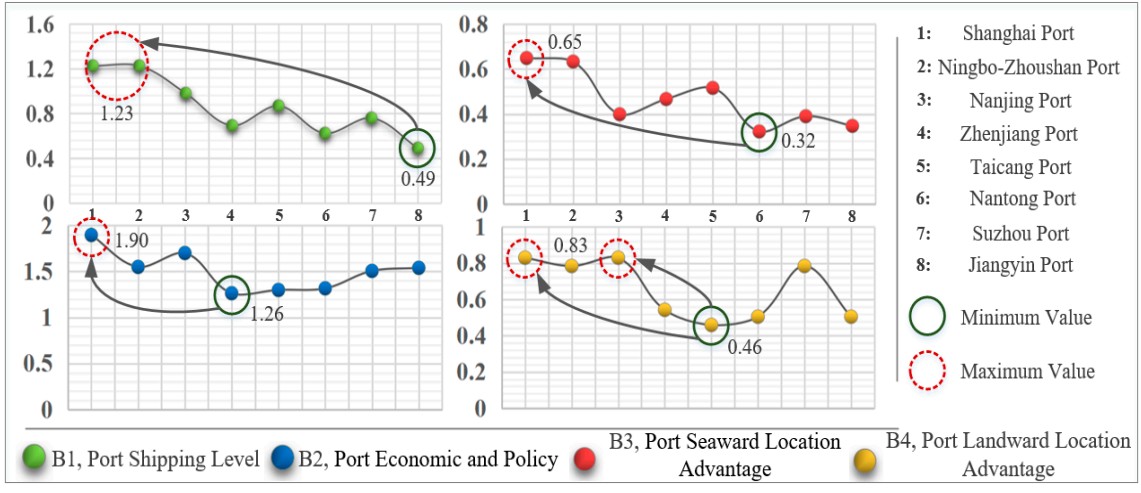

**Figure 4.** Score of each port in terms of the primary indicators.

From the perspective of port shipping level, the ranking order is as follows: Shanghai Port and Ningbo-Zhoushan Port, Nanjing Port, Taicang Port, Suzhou Port, Zhenjiang Port, Nantong Port, Jiangyin Port. And the index weights in the shipping level are ranked as number of berths (0.459), number of routes (0.362), port throughput (0.179). It can be seen that if the port wants to improve its own shipping level, more attention should be paid to increasing the number of routes and port throughput, and constructing dock berths for more ships to call, load and unload, etc., to avoid waiting in line so as to improve the overall operational efficiency of the port.

From the perspective of port economic and policy, the ranking is as follows: Shanghai Port, Nanjing Port, Ningbo-Zhoushan Port, Jiangyin Port, Suzhou Port, Nantong Port, Taicang Port, Zhenjiang Port. And the index weights in the economic and policy are ranked as per capita GDP (0.408), government support (0.167), attracting investment amount (0.161), environmental initiatives (0.131), and GDP growth rate (0.131), which shows that if the port wants to improve its own economic and policy indicators, more attention should be paid to the improvement of the overall living standards of the people in the hinterland. That is because per capita GDP reflects the level of social development in the region and the ability to invest in social construction, and the national income situation is also a condition that international investors pay great attention to, which indirectly affects port investment and regional economic development.

From the perspective of seaward location advantage, the ranking is as follows: Shanghai Port, Ningbo-Zhoushan Port, Taicang Port, Zhenjiang Port, Nanjing Port, Suzhou Port, Jiangyin Port, Nantong Port. And the index weights in the seaward location advantage are ranked as port network status (0.695), port strategic hub influence (0.186), and port strategic channel influence (0.119), which

shows that if the port wants to improve its seaward location advantage, more attention should be paid to increasing port throughput and expanding the scope of trade radiation. To increase port throughput is to expand port operations, the critical factor of which is to increase the berth capacity of the terminal, thereby driving the increase in foreign trade in the region.

From the perspective of landward location advantage, the ranking is as follows: Shanghai Port and Nanjing Port, Ningbo-Zhoushan Port and Suzhou Port, Zhenjiang Port, Nantong Port and Jiangyin Port, Taicang Port. The landward location advantage refers to the traffic accessibility of the hinterland and the support capacity of the economic development level for port development. The weights of the landward location advantage are ranked as road network density (0.425), traffic trunk influence (0.342), and urban influence (0.233), which shows that traffic conditions in the hinterland have a greater impact on the landward location advantage of the port. A perfect transportation network is the basis for the port to expand the hinterland, and the necessary condition for realizing rapid distribution of materials, which directly influences the operational efficiency of the port. Therefore, if the port wants to have a better development prospect, much should be invested in the construction of hinterland transportation network.

In addition, the rankings of the eight ports under the 14 secondary indicators (C1–C14) was obtained, as is shown in Table 9.

**Table 9.** The membership of each port in the secondary indicators.

| C1–C14 | Shanghai Port | Ningbo-Zhoushan Port | Nanjing Port | Zhenjiang Port | Taicang Port | Nantong Port | Suzhou Port | Jiangyin Port |
|--------|--------------|---------------------|--------------|----------------|--------------|--------------|-------------|---------------|
| C1 PT | 5 | 5 | 4 | 2 | 4 | 3 | 3 | 2 |
| C2 NR | 5 | 5 | 4 | 3 | 4 | 3 | 2 | 2 |
| C3 NB | 5 | 5 | 4 | 3 | 3 | 2 | 4 | 2 |
| C4 GDPGR | 4 | 4 | 4 | 3 | 2 | 4 | 2 | 3 |
| C5 PCGDP | 4 | 2 | 4 | 3 | 2 | 3 | 4 | 5 |
| C6 AIA | 5 | 5 | 4 | 2 | 4 | 3 | 3 | 2 |
| C7 GS | 5 | 5 | 4 | 3 | 4 | 2 | 3 | 2 |
| C8 EI | 5 | 5 | 4 | 4 | 5 | 4 | 5 | 4 |
| C9 SHI | 5 | 5 | 3 | 2 | 4 | 3 | 2 | 4 |
| C10 SCI | 4 | 3 | 3 | 3 | 3 | 4 | 4 | 4 |
| C11 NS | 5 | 5 | 3 | 4 | 4 | 2 | 3 | 2 |
| C12 UI | 5 | 4 | 5 | 2 | 2 | 3 | 4 | 3 |
| C13 TTI | 4 | 4 | 4 | 3 | 3 | 3 | 4 | 3 |
| C14 RND | 4 | 4 | 4 | 3 | 2 | 2 | 4 | 2 |

By analyzing the Table 9, the advantages and disadvantages of each port can be visually analyzed. The above research shows that economic and policy (B2) is the most important primary indicator affecting the evaluation of port development potential, and per capita GDP (C5) is the most important secondary indicator in port economic and policy evaluation. As can be seen from Table 9, Shanghai Port is at a relatively high level in terms of GDP growth rate, per capita GDP, attracting investment amount, government support, and environmental initiatives indicators, which are related to its superior geographical position, developed economic conditions and transportation system, foreign trade policy (which had an early start and is open), and green environmental protection concept of sustainable development. So, the development potential of Shanghai Port is greater than that of other ports. Except for Shanghai Port, the indicators of Ningbo-Zhoushan Port are higher than other ports except for per capita GDP. Its development potential is second only to Shanghai Port among the eight ports, mainly due to the fact that Ningbo-Zhoushan Port has fully utilized its deep-water advantages in recent years, grasping the pulse of the large-scale development of ships, promoting the berths of docks, upgrading facilities, and striving to improve the berthing capacity of super-large vessels, while always adhering to the green environmental protection mode to promote the joint operation and development of container railway and water transport. Although Nanjing Port enjoys only a medium level of attracting investment amount and government support indicators, its GDP growth rate and per capita GDP are equal to that of Shanghai Port. Therefore, Nanjing Port has the third highest

development potential with its deep water channels, wide port area, and good natural conditions. If the government gives sufficient policy support and financial support, on the premise of adhering to green and sustainable development, its development potential will be noticeable. Zhenjiang Port, Taicang Port, Nantong Port, and Suzhou Port are all at a similar level in terms of GDP growth rate, attracting investment amount, government support, and environmental initiatives, indicating that their development potential is mainly limited by per capita GDP indicators. In terms of per capita GDP, Suzhou Port is the highest, Taicang Port the lowest, and Zhenjiang Port and Nantong Port are medium with similar development potentials. Therefore, the development potential of Suzhou Port should be the largest, that of Zhenjiang Port and Nantong Port should be similar, and that of Taicang Port should be the smallest. It is worth noting that the development potential of Taicang Port is not the lowest; instead it is still higher than Zhenjiang Port and Nantong Port. The reason may be related to government support. Taicang Port is the only good natural port in Jiangsu Province—rare in China and rare in the world. The state and the Jiangsu's provincial government attach great importance to Taicang Port and have repeatedly pointed out that the port has a unique location advantage strategically, and must, therefore, be a focus of development acceleration. There is an interesting phenomenon in Jiangyin Port; that is, although the per capita GDP is high, the environmental initiatives are very low, and the ultimate development potential is also very low. The reason lies in the industrial agglomeration of electricity industry, chemical industry, and manufacturing industry in the hinterland of Jiangyin Port, which is a powerful industrial town. In recent years, environmental protection may have been ignored in pursuit of economic benefits. It can be seen that if a port does not pay attention to the concept of sustainable development, then no matter how strong the economy is, it cannot achieve good development in the future.

As is shown in Appendix A (Figure A1), the black curve indicates the degree of membership under the balanced development of various indicators of the port (the contribution of each indicator to the total development potential is equal), and the red curve indicates the degree of membership in actual development of the port indicators. It was found that, as the development potential of the port decreases, the black curve and the red curve became more and more similar. This shows that the indicators with large development potential are less balanced, with usually one or more than one outstanding performance. For example, Ningbo-Zhoushan Port, with a unique natural deep-water port, is currently superior to other ports in hardware facilities, but is not prominent in terms of software facilities such as informationization level, service quality, and collection and distribution systems. The development of various indicators in the port is relatively balanced, with no particularly outstanding indicator, such as in Zhenjiang Port and Nantong Port. In fact, it is very difficult for a port to achieve comprehensive, balanced, and rapid development. As we all know, everything in the world has its strong and weak points. Therefore, for ports with greater development potential, they should continue to vigorously carry forward their strengths and properly compensate for their shortcomings. For ports with average or poor development potential, full play should be focused on their comparative advantages, identifying the balancing points and breakthrough points of development, stimulating potential, and promoting development.

### 5.3. Discussion

"Port economy and policy" is the most important indicator with which to determine the development potential of a port, which shows that external conditions are more important than intrinsic conditions. That is to say, if a port has sufficient funds and sufficient government support, it will have a good development potential. The research on Singapore Port by Gordon et al. [60] shows that government support policies, sufficient investment funds, and well-designed operation can help the port to create sustainable development advantages. Therefore, all ports should optimize resource allocation, increase economic efficiency and investment quantity, and enhance environmental protection, so that their ports can develop healthily and sustainably. From the perspective of basic indicators, the three indicators of per capita GDP, number of berths, and sport network status, have

large impacts on the development potential of the port, playing a decisive role in the evaluation results. Among those, the impact of per capita GDP is the biggest, which shows that the investment in the construction of a port with high per capita income is most likely to reduce the investment risk and obtain greater returns. Therefore, if the port wants to have greater development potential, the living standards of the people in the hinterland must reach a certain height. However, as the case of Jiangyin Port, if the increase of per capita GDP is at the cost of environmental pollution and ecological damage, then the potential and possibility of the port to achieve better development in the future will be very low. Then, the number of berths is the second factor affecting the development potential of the port. It directly determines the number of ships that can be docked at the port terminal. It is the main parameter for verifying the port's capacity. Regional integration and globalization have brought more development to the ports. Venturini et al. have proven that with other conditions unchanged, properly increasing the port capacity can reduce the environmental pollution and energy waste caused by ships waiting in line, so as to attract more transport ships to berth [61], while Park et al. discussed the coping capacity of existing container terminals, and found that in order to allocate appropriate berths, large container ships have longer berthing times at the wharf. Therefore, the wharf should make differentiated berthing plans according to the size of the ship, berthing time, and cargo handling priority [62]. Port network status is the third factor affecting the development potential of ports, which matches the research results in reference [17]. Ports with high network status are mostly located in important waters, with strong port connectivity and wide radiation ranges, so often contain great development potential. It is therefore suggested that all ports should actively realize tax preference, relax investment restrictions, open trade cooperation, and improve their position in the maritime network by widening trade links and improving the route network.

　　　Ports with larger development potential usually have one or more outstanding indicators. For ports that have not found a breakthrough in development, that may be due to their unclear comparative advantages with other ports and their inaccurate positioning in the future national port planning, which has led to the backwardness of said ports' development potentials. In terms of the current state of development of each port, the relative advantage of Shanghai Port lies in the port's shipping level, and the relative disadvantage lies in the port's seaward location advantage; the relative advantage of Ningbo-Zhoushan Port lies in the port shipping level, and the relative disadvantage lies in the port's economy and policy; the relative advantage of Nanjing Port lies in the port's landward location advantage, and the relative disadvantage lies in the port's seaward location advantage; the relative advantage of Zhenjiang Port lies in the port seaward location advantage, and the relative disadvantage lies in the port's economy and policy; the relative advantage of Taicang Port lies in the port's seaward location advantage, and the relative disadvantage lies in the port's landward location advantage; the relative advantage of Nantong Port lies in the port's economy and policy, and the relative disadvantage lies in the port's seaward location advantage; the relative advantage of Suzhou Port lies in the port's landward location advantage, and the relative disadvantage lies in the port's seaward location advantage; the relative advantage of Jiangyin Port lies in the port's economy and policy, and the relative disadvantage lies in the port's shipping level. Therefore, in order to improve their development potential, each port should give full play to their relative advantages and make up for their relative disadvantages. First of all, it is necessary to analyze the development trend of the port shipping industry in the future, combining their own advantages to target the future needs of the world, the country, and the surrounding areas, and to seize the key links in the shipping network. Secondly, establish the concept of green development. Green development is a trend of globalization, which is conducive to port "high-quality" and "high-level" development. From the perspective of sustainable development, port management should balance the three aspects—economic prosperity, social welfare, and environmental quality [63]. In the ESPO (European Sea Ports Organisation) Green Guide, Shanghai Port, located in the Yangtze River Delta in China, has passed the green port assessment and is considered to have an excellent port environment and sustainable development ability [64]. From the perspective of policy and management, the ports' pursuits of high-quality and high-level

sustainable development require the government to skillfully combine ecological protection and economic returns to achieve green environmental protection in the shortest time [65]. Finally, it is recommended that governments at all levels should open up the investment field, and adopt effective policy incentives. In addition, the Matthew effect [66,67] shows that the leading places will take the lead, while the more backward places will lag behind, which explains why people usually invest in leading ports, rather than in backward ports. In order to improve this phenomenon, for leading ports, governments should give full play to the exemplary role of leading ports and offer help and support to backward ports; for backward ports, the state can give appropriate policy support, assisting with funds, projects, etc., to stimulate their internal initiatives.

With the development of society, there will be more and more factors affecting the development potential of each port. How to calculate the development potential of other ports and even global ports more accurately and conveniently is the next important issue to be explored.

## 6. Conclusions

This paper comprehensively considered important factors, such as port shipping level, port economic and policy, port seaward location advantage, and port landward location advantage, evaluated the development potentials of eight representative ports in the Yangtze River Delta port group through FAHP-entropy model, provided comments that can be used as a frame of reference for other ports, and explored the nature and law of port development potential, with significance for port investment and construction, regional trade cooperation, and sustainable development. The results were as follows:

1. Overall, the development potential of the port group in the Yangtze River Delta is relatively large. The specific ranking is: Shanghai Port (perfect), Ningbo-Zhoushan Port (perfect), Nanjing Port (good), Suzhou Port (good), Taicang Port (good), Nantong Port (average), Zhenjiang Port (average), Jiangyin Port (average).

2. From the perspective of the single indicator, the ranking of the impact of the four primary indicators on the port development potential is: port economic and policy, port shipping level, landward location advantage, and seaward location advantage, indicating that external conditions of ports are more important than their internal conditions. As long as sufficient capital investment and government support are given, each port is likely to have strong potential for development. The per capita GDP is the main factor determining a port's economy and policies. The number of berths is the main factor determining the port shipping level. The road network density is the main factor determining the port landward location advantage. The port network status is the main factor determining port seaward location advantage. The top three secondary indicators on a port's development potential are per capita GDP, number of berths, and port network status. If the three indicators of the port are more prominent, the development potential of the port is also bigger.

3. From the perspective of port development, if the port wants to have greater development potential, it must break the balanced development of various indicators, correctly analyze and fully grasp its own comparative advantages, avoid relative disadvantages, and invest in new technologies, making new ideas and new models driving forces for port development.

**Author Contributions:** All of the authors contributed to the work in the paper. Specifically: conceptualization, N.M. and L.Z.; data curation, C.W. and T.Y.; investigation, N.M. and L.Z.; methodology, N.M. and C.W.; supervision, L.Z.; validation, C.W.; visualization, C.W.; writing—original draft, N.M. and C.W.; writing—review and editing, N.M., C.W., and T.Y. All authors have read and agreed to the published version of the manuscript.

**Funding:** This research was supported in part by the National Natural Science Foundation of China (41771476).

**Conflicts of Interest:** The authors declare no conflict of interest.

## Appendix A

**Table A1.** Results of four model compatibility degree tests.

|  | FAHP-Entropy | AHP-Entropy | FAHP | Entropy | Compatibility Degree |
|---|---|---|---|---|---|
| FAHP-Entropy | 1.000 | 0.417 ** ($p < 0.001$) | 0.350 ** ($p < 0.001$) | 0.671 ** ($p < 0.001$) | 0.479 |
| AHP-Entropy |  | 1.000 | 0.489 ** ($p < 0.001$) | 0.238 ** ($p < 0.001$) | 0.381 |
| FAHP |  |  | 1.000 | 0.202 ** ($p < 0.001$) | 0.347 |
| Entropy |  |  |  | 1.000 | 0.370 |

Note: ** indicates significant at 1% confidence level.

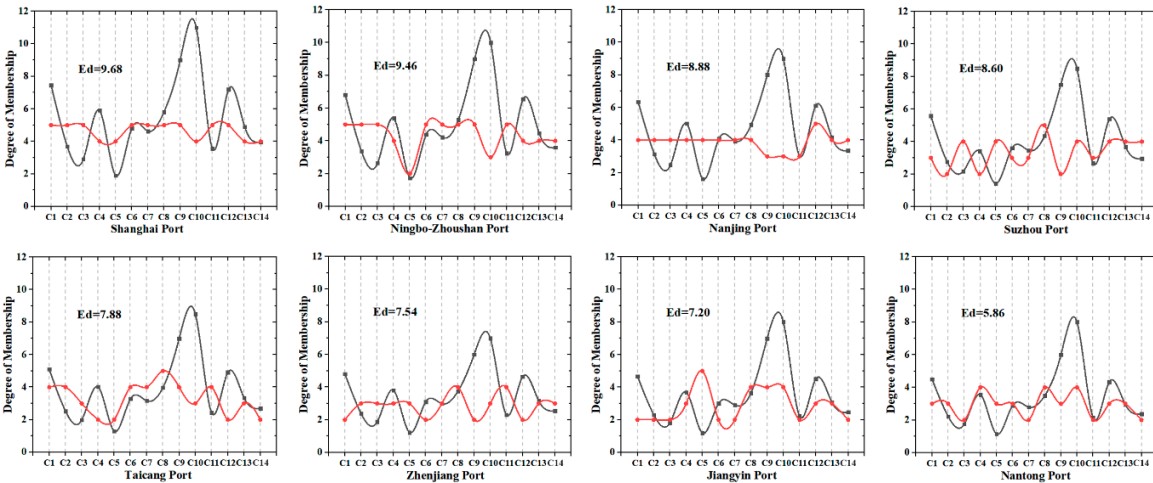

**Figure A1.** Analysis of port development status.

## Appendix B

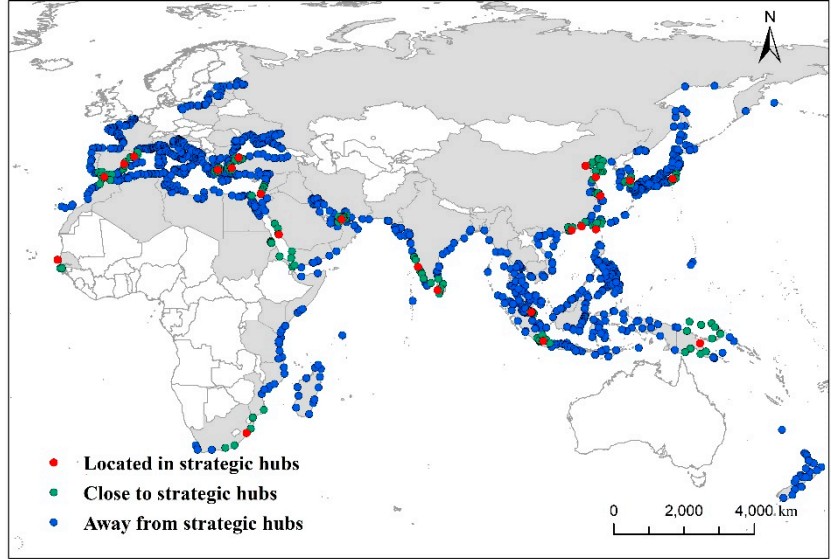

**Figure A2.** Spatial distribution of the strategic hub.

**Table A2.** The required data and processing steps.

| Indicators | Required Data and Data Sources | Processing Steps |
|---|---|---|
| C1 PT ($10^4$ ton) | Data: import amount + export amount<br>Source: provided by China Port<br>(http://www.port.org.cn) [52] | 1. Obtaining the official statistics of the total amount of various cargos<br>2. Normalization [a] |
| C2 NR (route number) | Data: the total number of routes in the study period<br>Source: AIS data in 2014, 2015 is provided by HiFleet Ltd. (http://www.hifleet.com) [50] | 1. Counting the number of port routes from OD data<br>2. Normalization |
| C3 NB (berth number) | Data: total number of berths<br>Source: provided by China Port<br>(http://www.port.org.cn) [52] | 1. Obtaining the official statistics of the total amount of various cargos<br>2. Normalization |
| C4 GDPG (%) | Data: total GDP of port cities<br>Source: provided by National data<br>(http://data.stats.gov.cn/) [48] | 1. Obtaining the total GDP of the port hinterland at the beginning and end of the study period<br>2. Substituting into the Equation (1)<br>3. Normalization |
| C5 CGDP (CNY [b]) | Data: total GDP and population of port cities<br>Source: provided by National data<br>(http://data.stats.gov.cn/) [48] | 1. Obtaining the total GDP and population of the port hinterland at the beginning and end of the study period<br>2. Total GDP divided by population<br>3. Normalization |
| C6 AIA ($10^4$ CNY) | Data: amount of investment attraction<br>Source: provided by National data<br>(http://data.stats.gov.cn/) [48] and port annual report [49] | 1. Obtaining investment funds of the port received during the study period<br>2. Normalization |
| C7 GS | Data: economic support and policy reference provided by the government<br>Source: provided by China Port<br>(http://www.port.org.cn) [52] and provincial statistical bulletin in 2014, 2015 [53] | 1. Assigning value by customizing the grading scale of indicators<br>2. Normalization |
| C8 EI | Data: port environmental protection policy issued by the government<br>Source: provided by China Port<br>(http://www.port.org.cn) [52] and provincial statistical bulletin in 2014, 2015 [53] | 1. Assignment by customizing the grading scale of indicators<br>2. Normalization |
| C9 SHI | Data: the shortest distance between port and hub port<br>Source: provided by World Port Index<br>(http://www.nga.mil/) [54] | 1. Reference [55], strategic hub distribution shown in Figure A2, using ArcGIS for shortest path analysis<br>2. Substituting into the Equation (2), where $\omega_j$ represents the weight of strategic hub level. The world's major ports are divided into five levels of shipping strategic hubs. In this study, the closest hub ports are Shanghai Port (0.9) and Qingdao Port (0.7).<br>3. Normalization |
| C10 SCI | Data: distance between the port and strategic passage<br>Source: provided by World Port Index<br>(http://www.nga.mil/) [54] | 1. Reference [55], strategic passage distribution shown in Figure A3<br>2. Assignment according to Table 4<br>3. Normalization |
| C11 NS | Data: Degree index, node number and node tightness<br>Source: provided by HiFleet Ltd.<br>(http://www.hifleet.com) [50] | 1. Calculating the degree index, node number and node tightness of each port (Equations (3)–(5))<br>2. Weighted calculation (Equation (6))<br>3. Normalization |
| C12 UI | Data: GDP of the port hinterland, the shortest distance between ports and cities<br>Source: provided by National data<br>(http://data.stats.gov.cn/) [48] and OpenStreetMap<br>(http://www.openstreetmap.org/) [51] | 1. Using ArcGIS to calculate the shortest distance between ports and cities<br>2. Obtaining GDP of the port hinterland<br>3. Using the spatial location potential model and substituting into the Equation (7).<br>4. Normalization |
| C13 TTI | Data: the shortest distance between ports and roads<br>Source: provided by OpenStreetMap<br>(http://www.openstreetmap.org/) [51] | 1. Using ArcGIS to calculate the shortest distance between ports and different roads<br>2. Substituting into the Equation (8), where $\omega_j$ represents the weight of different levels of roads, as shown in Table A3 (See reference [68,69])<br>3. Normalization |
| C14 RND (km/km$^2$) | Data: length of roads in the hinterland, hinterland area<br>Source: provided by National data<br>(http://data.stats.gov.cn/) [48] and OpenStreetMap<br>(http://www.openstreetmap.org/) [51] | 1. Calculating the hinterland area and total length of railways, highways and expressways in the hinterland.<br>2. substituting into the Equation (9), here $\omega_j$ represents the weight of different levels of roads, weight values are assigned to the following indexes: railway 0.4, expressway 0.3, and normal road 0.2. (See reference [70])<br>3. Normalization |

Note: [a] indicates the $C_i = (C_i - C_{min})/(C_{max} - C_{min})$; OD indicates origin–destination; [b] indicates the Chinese Yuan.

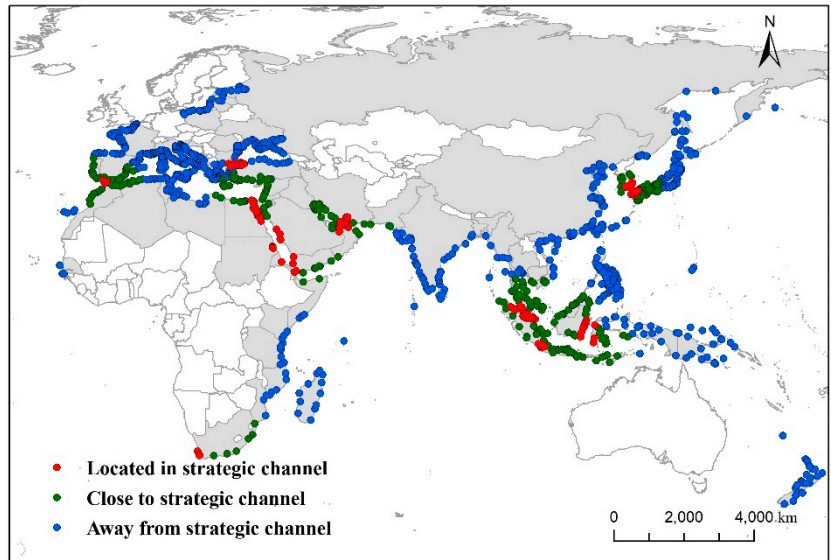

**Figure A3.** Spatial distribution of the strategic channel.

**Table A3.** Calculation parameters for traffic trunk influence.

| Respects | Range of Influence (km) | Attenuation Coefficient | Weights |
|---|---|---|---|
| Railway (railway station) | 60 | $-1/30$ | 2 |
| Highway | 45 | $-1/30$ | 1.5 |
| Trunk | 15 | $-1/15$ | 1 |
| Airport | 10 | $-1/10$ | 1 |

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
