# Peer review of "Evaluation of Development Potential of Ports in the Yangtze River Delta Using FAHP-Entropy Model"

_sustainability, doi:10.3390/su12020493_

Round 1

Reviewer 1 Report

Figure 5 is confusing, suggestion is to change it with table

Author Response

Dear Reviewer:

Thank you for your comments concerning our manuscript entitled “Evaluation of Development Potential of Ports in the Yangtze River Delta Using FAHP-Entropy Model” (ID: sustainability-673373). Those comments have the important guiding significance for revising and improving our manuscript. We have studied comments carefully and have made correction which we hope meet with approval. Revised portion are marked in red in the paper. Our responds to your comments are as flows:

Point 1: Figure 5 is confusing, suggestion is to change it with table.

Response 1: Thank you very much for your positive comments and valuable suggestions for our manuscript. We have changed the Figure 5 to a table (Table 9) as suggested (Line 520, page 17).

Once again, special thanks to you for your good comments!

Regards

Authors

Reviewer 2 Report

Many Development Potential studies on Ports have been done in recent years in China. While the combination of Entropy, AHP and Fuzzy may be/is original for port evaluation studies, the indicators used are quite common. It would have been very interesting to have environmental sustainability (policy) as part of the set of evaluation criteria, eventually balancing the ambition for growth, but unfortunately this is not the case. Environmental sustainability issues should have been raised at least under 'Third' point (line 523) in the Discussion, where  'high-quality' and 'high-level port development' are addressed.

For that reason, the paper better matches Transport Journals or Operations Research Journals.

Methodology: it can be questioned to what extent the methodology depends on 'subjective' decisions, influencing  the results. E.g.  which time periods involved in measuring GDP growth rate? idem Port throughput. How is governmental support measured and differentiated between the eight ports?

The supporting figures are too small and difficult to read.

Not clear is where GGR stands for Formula (1).

Minor:

-ref 1 is from 2009, while it refers to the 'at present' situation

-not sure that 'strategic fulcrum identification model' (line 262) is a common term,

Author Response

Dear Reviewer:

Thank you for your comments concerning our manuscript entitled “Evaluation of Development Potential of Ports in the Yangtze River Delta Using FAHP-Entropy Model” (ID: sustainability-673373). Those comments have the important guiding significance for revising and improving our manuscript. We have studied comments carefully and have made correction which used the "Track Changes" function in Microsoft Word. We have tried our best to revise our manuscript according to the your valuable comments and constructive suggestions, our responds to your comments are as flows:

Point 1: Many Development Potential studies on Ports have been done in recent years in China. While the combination of Entropy, AHP and Fuzzy may be/is original for port evaluation studies, the indicators used are quite common. It would have been very interesting to have environmental sustainability (policy) as part of the set of evaluation criteria, eventually balancing the ambition for growth, but unfortunately this is not the case. Environmental sustainability issues should have been raised at least under 'Third' point (line 523) in the Discussion, where 'high-quality' and 'high-level port development' are addressed.

For that reason, the paper better matches Transport Journals or Operations Research Journals.

Response 1: Your comments are all valuable and very helpful for revising and improving our manuscript. We appreciate your efforts in reviewing our manuscript. We have supplemented the environmental initiatives in our evaluation index system and explained the process of obtaining it and its grading standard in the revised manuscript (Lines 349, page 10). In addition, we have expanded the discussion part. In the third point of discussion, we referred to the relevant literature and tried our best to describe the important relationship between 'high quality' and 'high level port development' and environmental sustainability (green development) (Lines 655 to 663, page 21).

Point 2: Methodology: it can be questioned to what extent the methodology depends on 'subjective' decisions, influencing the results.

Response 2: Yes, there is a certain subjectivity in this method, but it is very small. We have tried our best to reduce its subjectivity and enhance its scientificity based on the traditional method.

First of all, our evaluation indicators include quantitative indicators and qualitative indicators. The scores of quantitative indicators are objective, while the scores of qualitative indicators are relatively subjective. For the scores of qualitative indicators, we have tried our best to make them more scientific. In this study, we have listed the evaluation standards of qualitative indicators through references and expert suggestions, as shown in Table 4 (Lines 349, page 10). Besides, we obtained the value or actual situation of each indicator of the port after consulting the relevant authoritative websites, and scored in strict accordance with the methods and steps, trying to give each indicator a convincing score.

Secondly, the FAHP-Entropy method used in this study is the most excellent and robust one among the commonly used AHP-Entropy, FAHP, Entropy and Fuzzy in terms of development potential, and it is also improved compared with the original FAHP-Entropy method. The construction of judgment matrix in the original AHP method and the construction of membership function in the original Fuzzy method only rely on human subjective consciousness and experience. Now, we determine the judgment matrix and membership function through subjective experience and objective value on the basis of Entropy weight and hierarchical single sort.

Thank you for pointing out. In the future we will further explore how to reduce the influence of subjective decisions on the results. Thank you for your valuable comment.

Point 3: E.g. which time periods involved in measuring GDP growth rate? idem Port throughput.

Response 3: Data is from January 1, 2014 to March 31, 2015. Due to the difficulties in obtaining AIS data, AIS data from January 1, 2014 to March 31, 2015 is used in this study to reflect the position and importance of ports in the complex maritime network, that is, the network status of ports. Therefore, in order to maintain the consistency, all indicators with year difference, such as port throughput, number of routes, number of specialized berths, GDP growth rate, are based on the data from January 1, 2014 to March 31, 2015. We have annotated the time period under the Table 3 in the revised manuscript (Lines 238, page 7). It needs to be mentioned that data obsolescence is our weakness. At present, we have used all the data we currently have for this study. We are also trying to find the latest data. If we have the updated data in the future, we will improve it.

Point 4: How is governmental support measured and differentiated between the eight ports?

Response 4: We are sorry for not explaining it clearly. In the revised version, we have added Table 4 (Lines 349, page 10), trying to explain the scoring standards of qualitative indicators in more detail.

Point 5: The supporting figures are too small and difficult to read.

Response 5: Thank you for pointing out. We have changed the supporting figures to an appropriate size.

Point 6: Not clear is where GGR stands for Formula (1).

Response 6: Thank you for pointing out. GGR stands for the capitalization of initials of GDP Growth Rate (C4) index, which is really inappropriate. The GGR has been revised to C4. In order to keep consistent, the same revisions have been made to other formulas (Lines 284 to 335, page 8 to 10).

Minor:

Point 7: Ref 1 is from 2009, while it refers to the 'at present' situation.

Response 7: In fact, more than 90% of the world’s trade volume has been completed by sea transportation [1] (Lines 36, page 1), thank you again for pointing out.

[1] Li, Z.; Xu, M.; Shi, Y. Centrality in global shipping network basing on worldwide shipping areas. GeoJ. 2015, 80, 47-60.

Point 8: Not sure that 'strategic fulcrum identification model' (line 262) is a common term.

Response 8: Duly noted. We corrected the name in Reference [2]. The ‘strategic fulcrum identification model’ is corrected as ‘identification model of global strategic shipping pivots’ (Lines 302, page 9).

[2] Wang, C.J.; Chen, P.R.; Chen, Y.H. The identification of global strategic shipping pivots and their spatial patterns. J. Geo. Sci. 2018, 28, 1215-1232.

Once again, special thanks to you for your constructive comments, and hope that the correction will meet with approval.

Regards

Authors

Reviewer 3 Report

Firstly, we must underline the great interest of the work done and the original approach.
Generally speaking, it seems that the starting point should be reformulated to suit the methodology and results.
It does not seem to us that the methodology addresses the development potential of ports, but rather their current state of development compared to each other, by assessing the characteristics that some ports must develop in order to achieve the development of others. There are no variables that allow us to assess the development potential of ports, but variables that explain their current development, external and internal to the port. Nothing tells us if ports have more potential to realize than the current, only that some have lower output, with lower inputs determined.
It seems to us that the issue of departure is not well posed and should be redirected to a comparative assessment of current development between ports and the relationship between the characteristics of each port and region and their current development, so that some ports can be gauged. they can reach the levels of the most developed ports? which is something quite different to look at the development potential of ports as a general model.
The introduction (lines 177 to 185) could be improved by improving all English in the paper.
Figure 2 needs to improve reading and clarify the steps to take so that it can be understandable.
It is essential to draw up a table with the various variables, the base authors, the type of sources and method of collection, dates, the scale, the treatment of the data of each variable and its transformation very clearly and simply.
All variables of lines 228 to 338 should be explained without further explanation, how they were determined, who and how they valued the subjective and objective variables, the amounts of data used and observations regarding the population, sources, the type of data and their transformation. It is very poorly transparent. It looks like a very cloudy area of ​​paper.
In table 5 it seems that the variable designations should be placed for a better understanding.
Line 370 - Results are not the best place to refer to the methodology.
Table 4 - It seems to us that the variables are not well selected and some may not be relevant to the intended effect.
Development potential cannot be gauged by current development, such as the current port cargo movement or the number of quays or shipping lines. The port may have many lines and cargo and may not have any further development potential. As such it can have very few lines and have a high development potential. Are different things.
Cargo port prodution will be an output variable of the current port development state and not an input variable of future port development.
How was it quantified and by whom was the value of the port economic and policy variables? They are very subjective variables and it is essential to understand which methodology is used for their measurement and collection. It is not clear.
Location variables are critical to the port's development potential, but how were they measured? How were they calculated and collected? What scales? Which authors used with these methodologies and starting questions?
Is the methodology best suited to the initial question about the port's development potential? Does the methodology used make it possible to draw conclusions on what each port must do to realize its potential? It seems unreliable to us, since many variables are not input for the future, but current output, and the methodology does not compare current and future outputs and inputs over time.
Figure 4 should be further explained by demonstrating its importance to the results and the analysis of the results.
We do not think that Figure 6 is of any interest and may be referred to the appendix.
The classification of variables and ports should not be done with the> symbol, but with plain text and written explanations.
Line 477, is the port of nogbo-Zhousha superior or has superior development potential? There seems to be some confusion in the conclusions derived from the bad starting questions.
Line 501, cannot conclude about ports that were not studied in this paper.
Line 548 to 552, the conclusions do not match the abstract and should be reconciled.

Author Response

Dear Reviewer:

Thank you for your comments concerning our manuscript entitled “Evaluation of Development Potential of Ports in the Yangtze River Delta Using FAHP-Entropy Model” (ID: sustainability-673373). Those comments have the important guiding significance for revising and improving our manuscript. We have studied comments carefully and have made correction which used the "Track Changes" function in Microsoft Word. We have tried our best to revise our manuscript according to your valuable comments and constructive suggestions, our responds to your comments are as flows:

Point 1: Firstly, we must underline the great interest of the work done and the original approach.

Response 1: Thank you very much for your positive comments.

Point 2: Generally speaking, it seems that the starting point should be reformulated to suit the methodology and results.

It does not seem to us that the methodology addresses the development potential of ports, but rather their current state of development compared to each other, by assessing the characteristics that some ports must develop in order to achieve the development of others. There are no variables that allow us to assess the development potential of ports, but variables that explain their current development, external and internal to the port. Nothing tells us if ports have more potential to realize than the current, only that some have lower output, with lower inputs determined.

Response 2: Thank you for your suggestion. It's very important. Because of your suggestion, we found the shortcomings in the current work. In the past studies, the evaluation of development potential is based on the evaluation indicators such as the level of port infrastructure, hinterland economic conditions, government support policies and the competitive advantage of land and sea. However, we agree with your suggestion and also believe that these indicators may not accurately predict the future development of the port. Therefore, in the revised manuscript, we added the indicator of environmental sustainability (policy) of the port, trying to reflect the development potential of the port, at the same time, on the theme of the paper, the revised version tends to compare the current state of development of the port, so as to provide opinions and suggestions for the development potential of the port.

Point 3: It seems to us that the issue of departure is not well posed and should be redirected to a comparative assessment of current development between ports and the relationship between the characteristics of each port and region and their current development, so that some ports can be gauged. they can reach the levels of the most developed ports? which is something quite different to look at the development potential of ports as a general model.

Response 3: Thank you for your valuable comment. We have redirected to a comparative assessment of current development between ports in the results,and the relationship between the characteristics of each port and region and their current development, Only through effective comparison can we make clear the gap of each port, and combine the weight of each index to indicate which aspect of the port should be developed first to improve its development potential as soon as possible, so as to make up for its disadvantages accurately and give full play to its advantages. In addition, convenient transportation, developed economy and favorable policies, especially the enhancement of the government's environmental awareness, provide a guarantee for the future development of the port. We believe that the key to reach the levels of the most developed ports is to recognize their own relative advantages, find a breakthrough point for development, and the ports with great development potential are most likely to catch up with the most developed ports in the near future.

Point 4: The introduction (lines 177 to 185) could be improved by improving all English in the paper.

Response 4: Finished. We’re sorry we didn't explain it clearly. We have been working hard to learn English. Now we believe the revised paper will provide a more readable description on the method and the main results of this study (Lines 205 to 213, page 5 to 6).

Point 5: Figure 2 needs to improve reading and clarify the steps to take so that it can be understandable.

Response 5: Thank you for pointing out that we have made change in the Figure 2. The figure presents the method framework on port development potential evaluation, which consists of eight parts:

Step 1. Determining the evaluation methods.

Step 2. Determining the evaluation index system.

Step 3. Determining the score of each port on each index, and calculating the weight of each index by Entropy method.

Step 4. Using Entropy weight to help construct the judgment matrix of AHP.

Step 5. Using judgment matrix to obtain the results of hierarchical single sort of each index and each port.

Step 6. Using again the Entropy weight and results of hierarchical single sort to construct the membership function of each port to the hierarchical fuzzy subset.

Step 7. Using a weighted average M (·,⊕) fuzzy composition operator, membership function and Entropy weight to obtain the port development potential.

Step 8. Testing the compatibility degree of each weighting method.

We have made an additional attempt to clarify the figure in the text section relating to it.

Point 6: It is essential to draw up a table with the various variables, the base authors, the type of sources and method of collection, dates, the scale, the treatment of the data of each variable and its transformation very clearly and simply.

Response 6: We have added the main reference review table of port development potential (Lines 100, page 3), including the key issues of the paper, selected indicators, research scale and date.

Point 7: All variables of lines 228 to 338 should be explained without further explanation, how they were determined, who and how they valued the subjective and objective variables, the amounts of data used and observations regarding the population, sources, the type of data and their transformation. It is very poorly transparent. It looks like a very cloudy area of paper.

Response 7: Thank you for your constructive comments We have revised our manuscript accordingly. We have made an attempt to further clarify this in the Appendix B section(Lines 717, page 23),we have further explained the data, data source and processing steps needed for each indicator. We have added the type of data and their transformation into an appendix to improve replicability.

Point 8: In table 5 it seems that the variable designations should be placed for a better understanding.

Response 8: Duly noted. We agree with you. We want to place the variable designations in Table 5, but the size of the table is limited. At the same time, we think the vertical table is more difficult to observe. Therefore, in the revised manuscript, we have to use the initialism instead (Lines 419, page 14).

Point 9: Line 370 - Results are not the best place to refer to the methodology.

Response 9: Thank you for your valuable comment. We carefully moved it to the Appendix A section (Lines 713, page 22), and discussed it in the relevant researches on evaluation methods of Literature Review (Lines 179 to 187, page 5).

Point 10: Table 4 - It seems to us that the variables are not well selected and some may not be relevant to the intended effect.

Response 10: Thank you for your suggestion. It's very important. Because of your suggestion, we found the shortcomings in the current work, and we will improve the scientific research level and make more achievements in the future work according to your suggestion.

Point 11: Development potential cannot be gauged by current development, such as the current port cargo movement or the number of quays or shipping lines. The port may have many lines and cargo and may not have any further development potential. As such it can have very few lines and have a high development potential. Are different things.

Cargo port prodution will be an output variable of the current port development state and not an input variable of future port development.

Response 11: Our point of view is as follows: potential mainly evaluates the port’s ability to grow and develop within the global maritime transport network, mainly represented by two aspects. The first, port economy and policy (ports are encouraged and supported by external economy and policy, which is related to human factors in the hinterland and easy to change). The second, location advantage (the resources surrounding the port and its potential ability to allocate those resources,not easy to change),is also the main indicator traditionally used by many scholars to evaluate port potential. Based on AIS data, this paper simulates the global maritime network, uses the transportation routes formed between ports in the context of economic globalization to reflect the position and importance of ports in China’s Yangtze River Delta in the maritime network from the two perspectives of seaward location advantage and landward location advantage, and tries to reveal the competitive advantages of ports in the sea and their potential to compete for the future market. Besides, the hinterland traffic situation is used to reflect the hinterland supporting capacity of the future development of ports in the Yangtze River Delta of China, trying to reveal the ports’ competitive advantages in hinterland and their potential to compete for the future market.

Point 12: How was it quantified and by whom was the value of the port economic and policy variables? They are very subjective variables and it is essential to understand which methodology is used for their measurement and collection. It is not clear.

Response 12: We’re sorry we didn't explain it clearly. For the five secondary indicators (C4-C8) in port economy and policy, among which C4 (GDP growth rate), C5 (Per capita GDP) and C6 (Attracting investment amount) are quantitative indicators, and the exact values can be found on authoritative websites, as shown in Table 3 (Lines 237, page 6), then what is needed is the normalization. C7 (Government support) and C8 (Environmental initiatives) are qualitative indicators. For the scores of qualitative indicators, we have tried our best to make them more scientific. In this study, we have listed the evaluation criteria of qualitative indicators through references and expert suggestions, as shown in Table 3 (Lines 349, page 10). After consulting many authoritative websites, we get the value or actual situation of each indicator of the port, and grade strictly according to the method and steps, trying to give each indicator a convincing score.

Point 13: Location variables are critical to the port's development potential, but how were they measured? How were they calculated and collected? What scales? Which authors used with these methodologies and starting questions?

Response 13: Port location is the primary condition to be considered for port planning and construction, and it is also an important factor representing the importance of ports and the connectivity of shipping network. Location advantage indicates the degree of spatial connection, formed upon geographic advantages, between the research objects and external objects[1], and it is a representation of location quantification. The geographical assessment of a geographical entity’s location advantages is often conducted from the following perspectives including infrastructure levels, economic policies, traffic convenience as well as the demand size of the geographical entity and the attractiveness of the attractive kernel. Currently, studies on location advantage mostly focus on commercial location choice [2], residential location choice [3,4], urban units [5] and tourist sites [6,7], etc.. and less have done on the location advantages of ports. Naixia Mou et al. [8] and Peng Peng et al. [9] conduct quantitative assessments on the location advantages of the important ports along the Maritime Silk Road, and they add Complex Network analysis and geographic location analysis to the traditional way of assessment.

This paper is concerned with both seaward and landward factors. Seaward assessment index includes the strategic hub influence, strategic channel influence and port network status, while landward assessment index involves urban influence, traffic trunk influence and hinterland road network density. Based on the indexes, the paper builds a comprehensive assessment model of ports’ location advantages to fully reveal the location advantages of ports in the Yangtze River Delta.

The detailed measurement method and data collection and calculation process are shown in Appendix B (Lines 717, page 23).

[1] Guo, J.K.; Wang, S.B.; Wang, H.; Wang, L.H. Comprehensive Measure of the Regional Advantages of National Scenic Area. Econ. Geogr. 2017, 37, 187-195.

[2] Holl, A.; Mariotti, I. The Geography of Logistics Firm Location: The Role of Accessibility. Networks Spat. Econ. 2018, 18, 337-361.

[3] Guidon, S.; Wicki, M.; Bernauer, T.; Axhausen, K. The social aspect of residential location choice: on the trade-off between proximity to social contacts and commuting. J. Transp. Geogr. 2019, 74, 333-340.

[4] Kryvobokov, M.; Bonnafous, A.; Bouf, D. Simulating Residential Location Choice at Different Geographical Scales: The Case of Lyon. Appl. Spat. Anal. Policy 2015, 8, 351-370.

[5] Wang, Z.; Han, Q.; de Vries, B. Land Use/Land Cover and Accessibility: Implications of the Correlations for Land Use and Transport Planning, Appl. Spat. Anal. Policy 2018, 1-18. Online First: https://doi.org/10.1007/s12061-018-9278-2.

[6] Venter, O.; Magrach, A.; Outram, N.; Klein, CJ.; Possingham, HP.; Di Marco, M.; Watson, JEM. Bias in protected-area location and its effects on long-term aspirations of biodiversity conventions. Conserv. Biol. 2018, 32, 127-134.

[7] Alrwajfah, M.M.; Almeida-García, F.; Cortés-Macías, R. Residents’Perceptions and Satisfaction toward Tourism Development: A Case Study of Petra Region, Jordan. Sustainability 2019, 11, 1907.

[8] Mou, N.X.; Liao, M.D.; Zhang, H.C.; Fu, X.; Peng, P.; Liu, X.L. Evaluation on location advantages of the ports along the Maritime Silk Road. J. Geo-Inf. Sci. 2018, 20, 613-622.

[9] Peng, P.; Yang, Y.; Lu, F.; Cheng, S.F.; Mou, N.X.; Yang, R. Modelling the competitiveness of the ports along the Maritime Silk Road with big data. Transp. Res. Part A Policy Pract. 2018, 118, 852-867.

Point 14: Is the methodology best suited to the initial question about the port's development potential? Does the methodology used make it possible to draw conclusions on what each port must do to realize its potential? It seems unreliable to us, since many variables are not input for the future, but current output, and the methodology does not compare current and future outputs and inputs over time.

Response 14: Your comments are all valuable and very helpful for revising and improving our manuscript,We have studied these comments carefully and have tried our best to revise our manuscript according to your valuable comments and constructive suggestions. First of all, at present, there is a lack of comprehensive and clear description of the evaluation index system of port development potential. Our key references have been listed below for you. On the basis of the current research, we have comprehensively considered port infrastructure, hinterland economy, hinterland traffic, policies and regulations, combined qualitative and quantitative indicators, selected 4 primary indicators including the port shipping level, port economy and policy, port seaward location advantage, and port landward location advantage, and 14 secondary indicators to constitute the evaluation index system, trying to reflect the development potential of the port. The environmental initiatives index is especially introduced, mainly including the port environmental protection management policies, which is closely related to the sustainable development of the port; secondly, in the discussion of the revised version, we added drawing conclusions on what each port must do to realize its potential, providing reference for ports in the Yangtze River Delta to achieve better development prospects.

References

Indicators

Methods

Area

Date

Gao et al. [10]

Government policy, investment trend, investment risk management

Fuzzy-AHP, ELECTRE III

Quanzhou Port

2018

Feng et al. [11]

The growth rate of GDP/ container throughput/ cargo throughput/ foreign trade import and export volume, port location advantage and port ecological environment

TOPSIS-

Entropy

Major container ports in China

2017

Wan et al. [12]

Port throughput/ GDP growth rate, government support, classification of inland waterways

AHP, D-S Reasoning

Wuhan Port

2018

Shanghai International Shipping Institute [13]

The growth rate of container throughput/ GDP, the investment amount attracted by the port, the number of new routes, the natural conditions of the port and the impact of government actions

Global container ports

2017

Pahl et al. [14]

Logistics demand and infrastructure

Arctic ports

2018

Burling et al. [15]

The import and export goods

Potential growth model

Hedland Port

2003

Peng et al. [16]

Density, trunk road, city scale, strategic channel, hub port, weighted degree centrality, weighted closeness centrality, weighted betweenness centrality

AHP-Entropy

99 ports along the Maritime Silk Road

2018

[10] Gao, T.; Na, S.; Dang, X.; Zhang, Y. Study of the competitiveness of Quanzhou Port on the Belt and Road in China based on a Fuzzy-AHP and ELECTRE III model. Sustainability 2018, 10, 1253.

[11] Feng, M.X.; Hu, J.K. Research on the port comprehensive strength evaluation and operation efficiency analysis based on the Entropy-TOPSIS and DEA method. J. Cent. China Norm. Univ. (Nat. Sci.) 2017, 51, 356-363.

[12] Wan, C.; Zhang, D.; Fang, H. Incorporating AHP and Evidential Reasoning for quantitative evaluation of inland port performance. Int. Ser. Oper. Res. Manag. Sci. 2018, 151-173.

[13] Shanghai International Shipping Institute. Global Port Development Report (2017). http://www.sisi-smu.org/. (accessed on 12 April 2018).

[14] Pahl, J.; Kaiser, B.A. Arctic port development. In Arctic Marine Resource Governance and Development.; Vestergaard, N., Kaiser, B., Fernandez, L., Nymand, Larsen J., Eds.; Springer Polar Sciences. Springer, Cham, 2018.

[15] Burling, M.; Hutton, I.; Schepis, J. et al. Planning for the Ultimate Development Potential of the Port of Port Hedland// Coasts & Ports 2003 Australasian Conference: Proceedings of the 16th Australasian Coastal and Ocean Engineering Conference, the 9th Australasian Port and Harbour Conference and the Annual New Zealand Coastal Society Confeience. Institution of Engineers, Australasian, 2003.

[16] Peng, P.; Yang, Y.; Lu, F.; Cheng, S.F.; Mou, N.X.; Yang, R. Modelling the competitiveness of the ports along the Maritime Silk Road with big data. Transp. Res. Part A Policy Pract. 2018, 118, 852-867.

Point 15: Figure 4 should be further explained by demonstrating its importance to the results and the analysis of the results.

Response 15: We are grateful for the suggestion. We further analyze the Figure 4 in the revised manuscript (Lines 426 to 466, page 15 to 16).

Point 16: We do not think that Figure 6 is of any interest and may be referred to the appendix.

Response 16: Thank you for pointing out, we have moved Figure 6 from the conclusions to the appendix A section as suggested.

Point 17: The classification of variables and ports should not be done with the> symbol, but with plain text and written explanations.

Response 17: Thank you for pointing out, we corrected.

Point 18: Line 477, is the port of nogbo-Zhousha superior or has superior development potential? There seems to be some confusion in the conclusions derived from the bad starting questions.

Response 18: We’re sorry we didn't explain it clearly. This means that Ningbo-Zhoushan port is superior to other ports in terms of hardware facilities. Whether it is superior in terms of development potential is not the point to be made here. What we want to express here is the law discovered by figure A1 in Appendix A (Lines 715, page 22), that is, “Ports with larger development potential usually have one or more than one outstanding indicator, while the potential of ports with balanced development among all indicators is relatively weak”. Therefore, taking Ningbo-Zhoushan port as an example, we find that Ningbo-Zhoushan port has a unique and outstanding natural deep-water port, which is superior to other ports in terms of hardware facilities.

This paper hopes to reflect the development potential of the port in terms of its own conditions, explore the different potential and possibility of different indicators of the port to achieve better development in the future, and provide suggestions for the realization of greater development potential. In fact, ports with good development potential are often attached importance to and supported by economy or policy, or have prominent regional advantages under the international pattern. From the perspective of development, such ports can better grasp the development opportunities and provide sufficient growth momentum for the future development of ports. Therefore, we take eight typical ports of the Yangtze River Delta port group in 2014 as the research object, and evaluate these ports on the basis of 14 indicators including “Port throughput”, “Number of routes”, “Number of berths”, “GDP growth rate”, “Per capita GDP”, “Attracting investment amount”, “Government support”, “Environmental initiatives”, “Strategic hub influence”, “Strategic channel influence”, “Network status”, “Urban influence”, “Traffic trunk influence”, and “Road network density”, in order to reflect the real port value and development potential, and provide important reference for the development of shipping industry. More importantly, according to your suggestion, the theme of the paper is inclined to drawing conclusions on what each port must do to realize its potential. Thank you again for your valuable suggestions.

Point 19: Line 501, cannot conclude about ports that were not studied in this paper.

Response 19: In the discussion of the revised version, we corrected the coverage of the conclusion and only made a comparative analysis of the development potential of eight ports in the Yangtze River Delta. And the development potential of other ports and even the global ports will be the next issue we need to focus on. Therefore, your comments are added as an interesting topic worthy of further discussion.

Point 20: Line 548 to 552, the conclusions do not match the abstract and should be reconciled.

Response 20: Duly noted. The conclusion and abstract have been coordinated.

Once again, special thanks to you for your constructive comments, and hope that the correction will meet with approval.

Regards

Authors

Round 2

Reviewer 2 Report

The paper is now more balanced: as requested, environmental sustainability is explicitly included.

There are various presentation issues (just examples):

line 142: start 4) at new line; is the numbering still OK? line 271: is the symbol OK? Figure 3: is text on environmental initiativeses congruent with this figure? Please check whole document. 

Author Response

Dear Reviewer:

Thank you for your comments concerning our manuscript entitled “Evaluation of Development Potential of Ports in the Yangtze River Delta Using FAHP-Entropy Model” (ID: sustainability-673373). Our paper has been substantially improved under your guidance and advice, especially the addition of environmental initiatives fill the paper with brandnew and exciting significance. We have studied your comments carefully and have made correction which used the "Track Changes" function in Microsoft Word. We hope this revision will receive your approval. Our responds to your comments are as follows:

Point 1: The paper is now more balanced: as requested, environmental sustainability is explicitly included.

Response 1: Thanks to your valuable advice, our paper has been significantly improved. Thank you very much for your recognition and praise.

There are various presentation issues (just examples):

Point 2: line 142: start 4) at new line; is the numbering still OK?

Response 2: Thank you for pointing out. We have reflected carefully on the proposal, concluding that our numbering was inappropriate and that the background of the study was not clearly presented to readers. Therefore, we correct the way of numbering and divide the research background into three categories, namely, 2.1. Relevant researches on evaluation indicators, 2.2. Relevant researches on evaluation data and 2.3. Relevant researches on evaluation methods. And we typeset (1), (2), (3), (4) in 2.1 in the new line (Line 101 to 166, page 3 to 4), which is also neat and consistent with the other numbering format of the full text. Finally, we check the numbers of the full text and correct the inappropriateness one by one. For instance, in the abstract, we change 1), 2), 3) into (1), (2), (3) (Line 19 to 29, page 1).

Faithfully, thank you again for your correction.

Point 3: line 271: is the symbol OK?

Response 3: The purpose for giving abbreviation to each indicator is not only to help the reader better understand the table involving each indicator, but also to avoid word redundancy and to leave sufficient space for the presentation of other data in the table (e.g. Table 7 and Table 9). We carefully consider your suggestions of last time and this time, and think over how the most appropriate symbol for C1-C14 should be expressed. With reference [1], we finally correct each symbol (Line 272 to 340, page 10). For example, we change the abbreviation of C4, GDP growth rate to GDPGR, and change the abbreviation of C5and Per capita GDP to PCGDP, because GDP is the initials of Gross Domestic Product, which is an inseparable whole, and therefore may appear slightly longer than other C1-C14 indicators. Key references in [1] are shown below:

C1 Number of Berth (NB)

C2 Storage Capacity (SC)

C3 Quay Length (QL)

C4 Number of Cargol Handling Equipment (NCHE)

C5 Number of Tug and Barge (NTB)

C6 Port's Cargo Throughput (PCT)

C7 Coefficient of Storage Utilisation (CSU)

C8 Level of Informatisation (LI)

C9 Port Safety Management (PSM)

C10 Quality Management of Human Resources (QMHR)

C11 Customer Satisfaction (CS)

C12 Quantity of Port Fixed Asset (QPFA)

C13 Operating Revenueof Port (ORP)

C14 Total Profits (TP)

C15 Port Throughput Growth Rate (PTGR)

C16 GDP Growth Rate of port city (GDPGR)

C17 Government Support (GS)

C18 Classification of Inland Waterways (CIW)

[1] Wan, C.; Zhang, D.; Fang, H. Incorporating AHP and Evidential Reasoning for quantitative evaluation of inland port performance. Int. Ser. Oper. Res. Manag. Sci. 2018, 151-173.

Point 4: Figure 3: is text on environmental initiativeses congruent with this figure?

Response 4: Duly noted. We have made corresponding correction (Figure 3, Line 271, page 8).

Point 5: Please check whole document.  

Response 5: Thank you very much for your constructive suggestions. This time, we have carefully and repeatedly checked the full text, corrected the number, symbols and expressions in the article, and try to avoid the recurrence of presentation issues. We are sorry for our careless mistakes, and thank you very much for your correction and hard work. In the future we will learn from you, cultivating rigorous academic attitude and cautious scientific ideas, and we hope to learn more knowledge from you!

Once again, thank you for your correction and suggestiosn! 

Regards

Authors

Reviewer 3 Report

Dear Authors,

Thanks for the review work done, the paper seems to be able to be published after English  and  references final check by the publication Editor.

Author Response

Dear Reviewer:

Thank you very much for your approval of the revision of our manuscript entitled “Evaluation of Development Potential of Ports in the Yangtze River Delta Using FAHP-Entropy Model” (ID: sustainability-673373). Under your guidance and suggestions, our paper has been substantially improved, especially the addition of methodological steps and data processing details, which makes our paper more detailed and perfect. Our responds to your comments are as flowing:

Point 1: 

Dear Authors,

Thanks for the review work done, the paper seems to be able to be published after English and references final check by the publication Editor.

Response 1: Thank you very much for your valuable advice. We are very happy that you have accepted these revisions. We hope to learn more knowledge from you in the future.

Once again, special thanks to you for your good comments!

Regards

Authors